# DOMAIN GENERALIZATION VIA CONTENT FACTORS ISOLATION: A TWO-LEVEL LATENT VARIABLE MODELING APPROACH

## ABSTRACT

The purpose of domain generalization is to develop models that exhibit a higher degree of generality, meaning they perform better when evaluated on data coming from previously unseen distributions. Models obtained via traditional methods often cannot distinguish between label-specific and domain-related features in the latent space. To confront this difficulty, we propose formulating a novel data generation process using a latent variable model and postulating a partition of the latent space into content and style parts while allowing for statistical dependency to exist between them. In this model, the distribution of content factors associated with observations belonging to the same class depends on *only* the label corresponding to that class. In contrast, the distribution of style factors has an *additional* dependency on the domain variable. We derive constraints that suffice to recover the collection of content factors block-wise and the collection of style factors component-wise while guaranteeing the isolation of content factors. This allows us to produce a stable predictor solely relying on the latent content factors. Building upon these theoretical insights, we propose a practical and efficient algorithm for determining the latent variables under the variational auto-encoder framework. Our simulations with dependent latent variables produce results consistent with our theory, and real-world experiments show that our method outperforms the competitors.

## 1 INTRODUCTION

Traditional machine learning models achieve outstanding prediction performance by relying on the independently and identically distributed (iid) assumption, i.e., the training and testing data follow the same distribution (Vapnik, 1991). However, due to the possibility of distribution shifts between the training and the testing data, models trained under the iid assumption may not perform well in real-world situations. Alongside this, collecting data from some specific usage scenarios can be cost-prohibitive and, in some cases, infeasible, resulting in a significant obstacle to using these models in crucial applications where the underlying data distribution is uncertain (Wang et al., 2022). The purpose of domain generalization (DG) is to address this challenge (Blanchard et al., 2011), namely by producing a model which performs well on unseen but related data from distributions located in new environments (domains). With such a model, it is possible to achieve stable predictions in various applications (Jumper et al., 2021).

A prevalent strategy to DG involves the use of a shared predictor, with the objective of learning *domain invariant (content) representations* from observations across domains as input (Li et al., 2018c; Zhou et al., 2022; Ding et al., 2022). Traditionally, these representations are learned within a latent feature space and aligned across domains (Ganin et al., 2016; Li et al., 2018b). However, these methods fail to guarantee that domain-specific (style) spurious features can be effectively removed (Ding et al., 2022). As a result, the generalizability of the predictor is constrained (Muandet et al., 2013). Recent research has aimed to address this issue by utilizing causal structure to model the data generation process. Causal features of the label are leveraged for stable predictions based on invariant causal mechanisms (Schölkopf et al., 2012; Zhang et al., 2013; Magliacane et al., 2018). However, these methods (Peters et al., 2016; Bühlmann, 2020; Mitrovic et al., 2021) directly link the output label to a subset of observed covariates instead of latent features, which would be less suitable for applications involving sensory-level data due to conceptual limitations (Chalupka et al.,

2015; Atzmon et al., 2020). To circumvent this limitation, Sun et al. (2021) introduces a latent causal model named LaCIM to enable learning causal features within the latent space and establish sufficient conditions for their identifiability. However, the assumption of an informative label distribution of LaCIM (Sun et al., 2021, Theorem B.5) is often infeasible in practical applications, and the model cannot allow interdependence between non-causal features and the label. To remedy these shortcomings, we propose developing a novel data generation model (§ 3.1) that mitigates the need for stringent assumptions regarding label distribution.

In our model, we incorporate a novel two-level latent space comprising high-level and middle-level spaces namely, with each space being partitioned into content and style parts. We adopt a general assumption that the content $\hat{\mathbf{z}}_c$ and style $\hat{\mathbf{z}}_s$ in the high-level space follow a simple distribution, such as the multivariate normal distribution (Kingma & Welling, 2014; Higgins et al., 2017; Chen et al., 2018). Unfortunately, in the absence of additional assumptions, it is impossible to identify these latent variables from the observed data (Hyvärinen & Pajunen, 1999; Locatello et al., 2019). To remedy this situation, we further introduce a middle-level latent space inheriting the same partition structure as the high-level one. In the middle-level space, the content factors $\mathbf{z}_c$ are derived from $\hat{\mathbf{z}}_c$ through label-specific functions, resulting in a *domain-invariant* distribution of $\mathbf{z}_c$. The style factors $\mathbf{z}_s$ are generated from $\hat{\mathbf{z}}_s$ via label-domain-specific, component-wise monotonic functions. Subsequently, the concatenated latent factors $[\mathbf{z}_c, \mathbf{z}_s]$ produce observations via an invertible and smooth mixing function, which is shared across all domains. As per our model, distributional shifts of observations across domains arise from the variations of $\mathbf{z}_s$.

The key to achieving DG based on our model is *recovering the distribution of content factors and isolating them from style factors*. For this purpose, we assume that the prior of the style factors $\mathbf{z}_s$ belongs to a general exponential family, conditioned on the specific label and domain, following (Khemakhem et al., 2020a). Additionally, we introduce a domain variability assumption to enforce sufficient changes in style factors across domains. With these assumptions, we can theoretically recover the content factors and isolate them from the style factors (Theorem 1 in § 4.3), building on the identifiability results of recent literature on nonlinear independent component analysis (ICA) (Khemakhem et al., 2020a;b) and content isolation (Von Kügelgen et al., 2021; Kong et al., 2022). Based on our theoretical discoveries, we utilize a variational auto-encoder (VAE) framework as outlined in (Khemakhem et al., 2020a) to recover the content factors and train a stable prediction model, which solely takes $\mathbf{z}_c$ as input (§ 5). We then demonstrate empirically that our method outperforms various advanced approaches on both synthetic and real-world data (§ 6).

**Contributions.** To summarize our main contributions:

- *A novel identifiable latent variable model:* We introduce a novel model that incorporates a two-level latent space (§ 3.1) to effectively depict data generation process and distributional shifts across different domains. By establishing sufficient conditions for identifying the distribution of latent factors (§ 4.3), we demonstrate the feasibility of achieving DG under this model (Prop. 1).

- *A practical learning approach:* We design a VAE-based learning algorithm to recover latent variables and ensure the isolation of content factors from style factors. Specifically, we train an invariant predictor on the content factors, which is applicable to all domains (§ 5).

- *Experiments:* We conduct experiments using synthetic and real-world datasets, including Colored `MNIST` (Arjovsky et al., 2019), `PACS`, and `Office-Home` to demonstrate the practical effectiveness of our theoretical approach to the disentanglement of latent variables and the isolation of content factors, and to give empirical support for the generality of our method (§ 6).

## 2 PROBLEM FORMULATION

**Formal setup.** Denote the input observation space as $\mathcal{X}$ and a specific feature vector as $\mathbf{x} \in \mathcal{X}$. Similarly, let $\mathcal{Y}$ denote the output label space and $\mathbf{y} \in \mathcal{Y}$ represent a label vector. We denote $D_i$ as a sample comprising $n_i$ iid realizations of the pair $(\mathbf{x}, \mathbf{y})$ following the distribution $P_{\mathbf{xy}}^{(i)}$ on $\mathcal{X} \times \mathcal{Y}$. We define $\mathcal{D}_{\mathrm{tr}}$ as the collection of all these samples, i.e. $\mathcal{D}_{\mathrm{tr}} = \{D_1, D_2, \cdots, D_{N_{\mathrm{tr}}}\}$, $\mathcal{E}_{\mathrm{tr}} = \{1, 2, \cdots, N_{\mathrm{tr}}\}$ as the domain index set, and $e \in \mathcal{E}_{\mathrm{tr}}$ as the domain variable. Then, let $P_{\mathbf{xy}}^T$ be the distribution of the test set along with its corresponding test sample $D_T = \{(\mathbf{x}^{(j)}, \mathbf{y}^{(j)})\}_{j=1}^{n_T}$, which are $n_T$ iid realizations from $P_{\mathbf{xy}}^T$ and the labels $\mathbf{y}^{(j)}$ are unobserved. In the context of domain generalization, we have access

to all data in $\mathcal{D}_{\text{tr}}$, yet lack any information regarding the sample $D_T$ or distribution $P^T_{\mathbf{xy}}$ specific to the testing domain.

**Average Risk.** Given a classification loss function $\ell : \mathbb{R} \times \mathcal{Y} \to \mathbb{R}^+$, the objective of domain generalization is to produce an estimate $\phi : \mathcal{X} \to \mathbb{R}$, which can generalize well to $D_T$ drawn from the testing distribution $P^T_{\mathbf{xy}}$ rather than fit the observations $\mathcal{D}_{\text{tr}}$ used for training. Formally, the learning objective is to minimize the average generalization error $\mathcal{R}$ of the learned estimate $\phi$ over the unseen data $D_T$ of size $n_T$ (Blanchard et al., 2011; Muandet et al., 2013):

$$\mathcal{R}(\phi) := \mathbb{E}_{D^T \sim (P^T_{\mathbf{xy}})^{\otimes n_T}} \left[ \frac{1}{n_T} \sum_{i=1}^{n_T} \ell(\phi(\mathbf{x}^{(i)}), \mathbf{y}^{(i)}) \right] \tag{1}$$

## 3 THE DATA GENERATION MODEL

To achieve DG defined by Eq. 1 in § 2, we start by introducing a novel data generation model as shown in Figure 1. This model builds upon the foundations laid by LaCIM (Sun et al., 2021) and iMSDA (Kong et al., 2022), partitioning the latent space into content and style components.

### 3.1 MODEL DEFINITION

**High-Level Latent Space:** We define our data generation model by starting from specifying the variables in the high-level latent space, denoted as $\hat{\mathbf{z}} \in \mathcal{Z} \subset \mathbb{R}^{n_c+n_s}$. Also, we assume that each variable is mutually independent. This space consists of two parts: the content part $\hat{\mathbf{z}}_c \in \mathcal{Z}_c \subset \mathbb{R}^{n_c}$ and the style part $\hat{\mathbf{z}}_s \in \mathcal{Z}_s \subset \mathbb{R}^{n_s}$. Conceptually, we assume that these variables contain *basic components* that can lead to the generation of any observation $\mathbf{x}$. In other words, we assume an implicit mapping function $\mathcal{Z} \to \mathcal{X}$ exists.

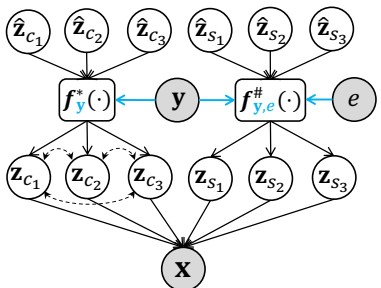

Justification. Dividing the latent space into two components and assuming variable independence are quite common in DG (Ganin et al., 2016; Sun et al., 2021; Lu et al., 2021; Kong et al., 2022; Taeb et al., 2022). For instance, in image generation, $\hat{\mathbf{z}}_c$ can capture the basic object concepts/components, while $\hat{\mathbf{z}}_s$ can encode the style/context information. *However, without additional assumptions, the latent variable $\hat{\mathbf{z}}$ is not identifiable from the observations $\mathbf{x}$* (Hyvärinen & Pajunen, 1999; Locatello et al., 2019), making it challenging to further isolate the content factors from the style factors. To address this issue, we propose introducing a middle-level latent space that enhances the level of detail in generating observations, thereby achieving model identifiability.

Figure 1: **The data generation model**. The gray-shaded nodes indicate that the variables are observable, and the blank nodes represent the latent variables during training. The dotted line with double arrows only implies the variables are allowed to be (conditionally) dependent. In this example, we have both $n_c$ and $n_s$ set to three.

**Middle-Level Latent Space:** We introduce a middle-level latent space and its variables, referred to as $\mathbf{z} \in \mathcal{Z} \subset \mathbb{R}^{n_c+n_s}$, which lives in the same space and retains the same division as the high-level latent space, with content factors $\mathbf{z}_c \in \mathcal{Z}_c \subset \mathbb{R}^{n_c}$ and style factors $\mathbf{z}_s \in \mathcal{Z}_s \subset \mathbb{R}^{n_s}$. In detail, $\mathbf{z}_c$ is derived from $\hat{\mathbf{z}}_c$ using non-linear functions denoted as $f^*_{\mathbf{y}}(\cdot)$. Here, the choice of $f^*$ depends on the value of $\mathbf{y}$ rather than taking it as an input. Similarly, $\mathbf{z}_s$ can be obtained through the application of component-wise monotonic functions $f^{\#}_{\mathbf{y},e}(\cdot)$ on $\hat{\mathbf{z}}_s$, which depends on (i) the values of the label $\mathbf{y}$, (ii) the domain variable $e$, (iii) and the index of each variable $i$. Our model goes beyond the previous assumptions (Sun et al., 2021) to a more general case by introducing dependence between the label and content factors and between the label and style factors (context information).

**Common Generation Principle:** The observation $\mathbf{x}$ is generated using a mixing function $g : \mathcal{Z} \to \mathcal{X}$ applied on the middle-level latent space $\mathbf{z} := [\mathbf{z}_c, \mathbf{z}_s]$, which can be written as follows:

$$\mathbf{x} = g(\mathbf{z}). \tag{2}$$

The mixing function $g$ is shared across all domains. To better understand this, let us examine the process of generating digits in the Colored MNIST dataset (Sun et al., 2021). The variations in the

joint distribution across domains of this dataset primarily stem from changes in the color style factor, which aligns with our model. For detailed explanation, refer to Example 1 in Appendix B.

## 3.2 PROBABILISTIC FORMULATION

**Density Transformation.** Formally, from Figure 1, we have the transformation between $\hat{\mathbf{z}}$ and $\mathbf{z}$ as

$$\hat{\mathbf{z}}_c \sim P(\hat{\mathbf{z}}_c); \quad \mathbf{z}_c = f_{\mathbf{y}}^*(\hat{\mathbf{z}}_c); \quad p(\mathbf{z}_c|\mathbf{y}) = \frac{p(\hat{\mathbf{z}}_c)}{|\det(\mathbf{J}(f_{\mathbf{y}}^*))|}, \tag{3}$$

$$\hat{\mathbf{z}}_s \sim P(\hat{\mathbf{z}}_s); \quad \mathbf{z}_s = f_{\mathbf{y},e}^\#(\hat{\mathbf{z}}_s); \quad p(\mathbf{z}_s|\mathbf{y},e) = \frac{p(\hat{\mathbf{z}}_s)}{|\det(\mathbf{J}(f_{\mathbf{y},e}^\#))|}, \tag{4}$$

where $\mathbf{J}(\cdot)$ represents the Jacobian matrix of the function and $\det(\cdot)$ calculates the matrix determinant.

**Probabilistic Formulation:** The probabilistic generative model can be easily formulated according to the model depicted in Figure 1 by

$$p_\theta(\mathbf{x},\mathbf{z}|\mathbf{y},e) = p_g(\mathbf{x}|\mathbf{z})p_{\boldsymbol{\theta}_c,\boldsymbol{\theta}_s}(\mathbf{z}|\mathbf{y},e), \tag{5}$$

where $\boldsymbol{\theta}_c$ and $\boldsymbol{\theta}_s$ control the distributions of $p(\mathbf{z}_c)$ and $p(\mathbf{z}_s)$ respectively and $\boldsymbol{\theta} = (g, \boldsymbol{\theta}_c, \boldsymbol{\theta}_s)$, living in the space $\Theta$, represents the parameters set of the model. (Notice that we ignore the high-level latent variable $\hat{\mathbf{z}}$ in Eq. 5 as the transformation between $\hat{\mathbf{z}}$ and $\mathbf{z}$ is deterministic.) And we also define:

$$p_g(\mathbf{x}|\mathbf{z}) = p_\epsilon(\mathbf{x} - g(\mathbf{z})). \tag{6}$$

Here, we assume an additive noise model $\mathbf{x} = g(\mathbf{z}) + \epsilon$, where $\epsilon$ is an independent noise variable with the pdf $p_\epsilon(\epsilon)$, following the definition in (Khemakhem et al., 2020a). Then, we can easily obtain the following conclusions based on the data generation model in Figure 1.

**Remark 1** *In our model, (a) $\mathbf{x} \perp\!\!\!\perp \mathbf{y}, e|\mathbf{z}$, meaning that observations are independent of the label and the domain variable given all latent factors $\mathbf{z}$; (b) $\mathbf{z}_c \perp\!\!\!\perp e$, meaning that $p(\mathbf{z}_c|\mathbf{y})$ keeps invariant across all domains.*

From Remark 1, we then can have the following proposition.

**Proposition 1** *Assume that we observe data sampled from a generative model defined according to Eqs. (3,4,5,6). Domain generalization can be achieved if a method can (1) recover the content factors $\mathbf{z}_c$ from the given observation $\mathbf{x}$ and ensure its isolation from the style factors $\mathbf{z}_s$; (2) learn an optimal invariant estimate $\phi : \mathcal{Z}_c \to \mathbb{R}$ only depending on $\mathbf{z}_c$.*

## 4 MODEL IDENTIFIABILITY

Based on the data generation model in Figure 1 and its probabilistic formulation presented in Eqs. (3,4,5,6), this section presents sufficient conditions for achieving the theoretical results on latent variable identifiability, which mainly include the recovery of $\mathbf{z}_c$ and its isolation from $\mathbf{z}_s$.

### 4.1 ASSUMPTIONS ON THE PRIOR

When the data generation model meets Eqs. (3,4,5,6), *the core idea to achieve the identifiability of the latent variables is enforcing their informative distribution*. Therefore, in our model, the prior on the latent variable $p(\mathbf{z}_s|\mathbf{y},e)$ is assumed to follow a general exponential family given by an arbitrary function $\boldsymbol{\lambda}_{\mathbf{z}_s}$ and sufficient statistics set $\mathbf{T}_{\mathbf{z}_s}$ concatenating the sufficient statistics of all variables in $\mathbf{z}_s$. No assumptions are made regarding the prior distribution of $\mathbf{z}_c$. Therefore, for simplicity, we use $\mathbf{T}$ and $\boldsymbol{\lambda}$ to represent $\mathbf{T}_{\mathbf{z}_s}$ and $\boldsymbol{\lambda}_{\mathbf{z}_s}$ throughout this paper. Then, the resulting probability density function is represented as

$$p_{\mathbf{T},\boldsymbol{\lambda}}(\mathbf{z}_s|\mathbf{y},e) = \prod_{i=1}^{n_s} \frac{Q_i(\mathbf{z}_{s_i})}{\Gamma_i(\mathbf{y},e)} \exp\left[\sum_{j=1}^k T_{ij}(\mathbf{z}_{s_i})\lambda_{ij}(\mathbf{y},e)\right], \tag{7}$$

where $\mathbf{Q} = [Q_i]$ is the base measure, $\boldsymbol{\Gamma} = [\Gamma_i]$ is the conditional normalizing constant, and $\mathbf{T}_i = (T_{i,1}, \cdots, T_{i,k})$ are the sufficient statistics of $\mathbf{z}_{s_i}$, and $\boldsymbol{\lambda}_i(\mathbf{y},e) = (\lambda_{i,1}(\mathbf{y},e), \cdots, \lambda_{i,k}(\mathbf{y},e))$ are the corresponding parameters depending on the values of $\mathbf{y}$ and $e$. As asserted in (Khemakhem et al., 2020a), exponential families possess universal approximation capabilities that enable the derivation of univariate conditional distributions of the latent sources, making this assumption nonrestrictive.

## 4.2 IDENTIFIABILITY DEFINITION

By making the conditional exponential family assumption for $\mathbf{z}_s$, we then introduce the block and the linear identifiability for latent factors $\mathbf{z}_c$ and $\mathbf{z}_s$, respectively.

**Definition 1** *[**Block identifiability.**] The content factors $\mathbf{z}_c$ are block-wise identifiable if the recovered $\mathbf{z}'_c$ contains **all and only** information about $\mathbf{z}_c$. That is to say, there exists an invertible function $h_c : \mathbb{R}^{n_c} \to \mathbb{R}^{n_c}$, that $\mathbf{z}'_c = h_c(\mathbf{z}_c)$.*

**Definition 2** *[**Linear identifiability.**] Let $\sim^L$ be the equivalence relations on $\{g, \boldsymbol{\theta}_s \coloneqq (\mathbf{T}, \boldsymbol{\lambda})\}$ defined as:*

$$g, \boldsymbol{\theta}_s \sim^L g', \boldsymbol{\theta}'_s \iff \exists\, A, \mathbf{b} \mid \mathbf{T}(g^{-1}(\mathbf{x})_{-n_s:}) = A\mathbf{T}'(g'^{-1}(\mathbf{x})_{-n_s:}) + \mathbf{b}, \tag{8}$$

*where $A \in \mathbb{R}^{kn_s \times kn_s}$ is an invertible matrix and $\mathbf{b} \in \mathbb{R}^{kn_s}$ is a vector. The subscript $-n_s:$ means extracting the last $n_s$ terms of a vector.*

Justification. Block identifiability, focusing on recovering variable blocks rather than each individual factor, is a well-studied topic in independent subspace analysis (Theis, 2006; Von Kügelgen et al., 2021). Linear identifiability (Hyvarinen & Morioka, 2016; Khemakhem et al., 2020a;b) states that latent factors' sufficient statistics can be identified up to an affine and component-wise transformation. Furthermore, under mild conditions from linear ICA (Hyvarinen & Morioka, 2016), the linear transformation in Definition 2 can be removed. As stated in Proposition 1, we aim to recover the $\mathbf{z}_c$ given observations in new domains. While it is not necessary to disentangle each individual factor but to ensure that all information in $\mathbf{z}_c$ is effectively recovered and contains no information from $\mathbf{z}_s$.

## 4.3 IDENTIFIABILITY RESULTS

**Theorem 1** *Assume we observe data sampled from the data generation model aligns with Eqs. (5,6,7), with parameters $(g, \mathbf{T}, \boldsymbol{\lambda})$, and the following assumptions hold:*

i. *[**Smooth and positive density**] The set $\{\mathbf{x} \in \mathcal{X} | \psi_{\boldsymbol{\epsilon}}(\mathbf{x}) = 0\}$ has measure zero where $\psi_{\boldsymbol{\epsilon}}$ is the characteristic function of the density $p_{\boldsymbol{\epsilon}}$. The probability density function of latent variables is smooth and positive, i.e., $p(\mathbf{z}|\mathbf{y}, e)$ is smooth and $p(\mathbf{z}|\mathbf{y}, e) > 0$ for all $\mathbf{z} \in \mathcal{Z}$, $\mathbf{y} \in \mathcal{Y}$, and $e \in \mathcal{E}_{\mathrm{tr}}$.*

ii. *[**Diffeomorphism**] Function $g$ in Eq. 6 is $\mathcal{D}^2$-diffeomorphism, i.e., it is injective, and all second-order cross-derivatives of the function and the inverse exist.*

iii. *[**Linear independence**] The sufficient statistics in $\mathbf{T}$ are all twice differentiable and $(T_{ij})_{1 \le j \le k}$ are linearly independent on any subset of $\mathcal{X}$ of measure greater than zero. Furthermore, they all satisfy $dim(\mathbf{T}_i) \ge 2, \forall i$; or $dim(\mathbf{T}_i) = 1$ and $\mathbf{T}_i$ is non-monotonic $\forall i$.*

iv. *[**Sufficient variability**] There exist $n_{\mathbf{y}}$ distinct $\mathbf{y}$ and each $\mathbf{y}^{(i)}$ locates in $n_{\mathbf{y}^{(i)}}$ distinct domains from $\mathcal{D}_{tr}$. There are $\sum_{i=1}^{n_{\mathbf{y}}} n_{\mathbf{y}^{(i)}}$ distinct points $(\mathbf{y}^{(i)}, e_i^{(j)})$ for $i \in \{1, \cdots, n_{\mathbf{y}}\}$, and $j \in \{1, \cdots, n_{\mathbf{y}_i}\}$ for each $i$ to have $\sum_{i=1}^{n_{\mathbf{y}}} n_{\mathbf{y}_i} \ge n_{\mathbf{y}} + kn_s$, and $\forall \mathbf{y}_i, n_{\mathbf{y}_i} \ge 2$. Assume that the matrix*

$$L = (\boldsymbol{\lambda}(\mathbf{y}^{(1)}, e_1^{(2)}) - \boldsymbol{\lambda}(\mathbf{y}^{(1)}, e_1^{(1)}), \cdots, \boldsymbol{\lambda}(\mathbf{y}^{(1)}, e_1^{(n_{\mathbf{y}^{(1)}})}) - \boldsymbol{\lambda}(\mathbf{y}^{(1)}, e_1^{(1)}), \cdots,$$
$$\boldsymbol{\lambda}(\mathbf{y}^{(n_{\mathbf{y}})}, e_{n_y}^{(2)}) - \boldsymbol{\lambda}(\mathbf{y}^{(n_{\mathbf{y}})}, e_{n_y}^{(1)}), \cdots, \boldsymbol{\lambda}(\mathbf{y}^{(n_{\mathbf{y}})}, e_{n_y}^{(n_{\mathbf{y}^{(n_{\mathbf{y}})}})}) - \boldsymbol{\lambda}(\mathbf{y}^{(n_{\mathbf{y}})}, e_{n_y}^{(1)})) \tag{9}$$

*of size $kn_s \times kn_s$ is invertible.*

v. *[**Domain variability**] For any set $\mathbb{A}_{\mathbf{z}} \subseteq \mathcal{Z}$ with the following two properties: (1) $\mathbb{A}_{\mathbf{z}}$ has nonzero probability measure, i.e., $\mathbb{P}[\{\mathbf{z} \in \mathbb{A}_{\mathbf{z}}\} | \{e = e'\}]$ for any $e' \in \mathcal{E}_{\mathrm{tr}}$. (2) $\mathbb{A}$ cannot be expressed as $\mathbb{B}_{\mathbf{z}_c} \times \mathcal{Z}_s$ for any set $\mathbb{B}_{\mathbf{z}_c} \subset \mathcal{Z}_s$. $\exists e_1, e_2 \in \mathcal{E}_{\mathrm{tr}}$, such that $\int_{\mathbf{z} \in \mathbb{A}_{\mathbf{z}}} P(\mathbf{z}|e_1) d\mathbf{z} \ne \int_{\mathbf{z} \in \mathbb{A}_{\mathbf{z}}} P(\mathbf{z}|e_2) d\mathbf{z}$.*

*Then, in the limit of infinite data, we can achieve the block identifiability of the content factors $\mathbf{z}_c$ and the linear identifiability of the style factors $\mathbf{z}_s$.*

Justification. The first three assumptions are trivial and easy to satisfy (Khemakhem et al., 2020b) (More detailed explanations are shown in the Appendix). **Linear independence**, coupled with the

**Sufficient variability**, constrains the space of sufficient statistics for $\mathbf{z}_s$, allowing for sufficient complexity of the latent sources distributions. Then, by aligning the marginal distribution of observations in each domain, it is possible to implicitly align the sufficient statistics of the style factors, further enabling the recovery of sufficient statistics under the invertible matrix assumption (Khemakhem et al., 2020a). Our assumption of sufficient variability is more practical as we do not restrict the label distribution to be informative like LaCIM (Sun et al., 2021) does. Moreover, it is relatively easier to satisfy our assumption compared to iMSDA (Kong et al., 2022) as the constraint of having an invertible matrix depends on both the label and domain variables in our case, whereas iMSDA only relies on the domain variable. The inclusion of **domain variability assumption**, introduced in (Kong et al., 2022), ensures significant variation in the distribution of $\mathbf{z}_s$ across domains, facilitating the isolation and recovery of content factors. The proof sketch, based on the contradiction of (iv) in Theorem 1, is presented in the Appendix. By showing that $\mathbf{z}_c'$ depends only on $\mathbf{z}_c$ and not $\mathbf{z}_s$, the proof ensures that invariance is not violated in any non-zero measure region the style subspace.

## 5 PROPOSED METHOD

In this section, we discuss how to turn the theoretical insights from § 4 into a practical algorithm, which can be used to achieve DG. As discussed in Proposition 1, the key is to block-wise identify the content factors $\mathbf{z}_c$ and train a stable predictor $\phi$ over the space of $\mathcal{Z}_c$. To achieve this, we then explain how to use a VAE framework (Kingma & Welling, 2014), in which the decoder neural network corresponds to the mixing function $g$, to estimate each latent component in the data generation model. The overall structure is illustrated in Figure 2.

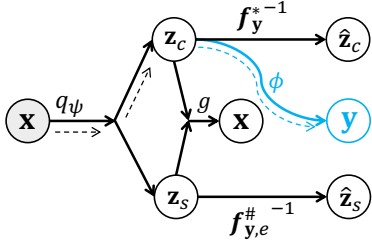

Figure 2: **The learning and inference procedure**. We take the encoder $q_\psi$ of a VAE to estimate the posterior of the latent factors, which is further used to (1) recover the high-level variable; (2) reconstruct the observations; (3) predict the label with the content factors. The solid line means the learning procedure and the dashed line denotes the inference procedure in new domains.

We consider estimating the latent variables by reformulating the objective function of VAE to fit each conditional marginal distribution $p(\mathbf{x}|\mathbf{y}, e)$. In vanilla VAE, a variational approximator $q_\psi(\mathbf{z}|\mathbf{x})$ is learned to approximate the intractable true posterior $p_\theta(\mathbf{z}|\mathbf{x})$ by maximizing the Evidence Lower Bound (ELBO). By utilizing Remark 1, our model similarly maximizes the conditional likelihood, formulated as

$$\mathbb{E}_{\mathcal{D}_{\text{tr}}}[\log p_\theta(\mathbf{x}|\mathbf{y}, e)] \geq \mathcal{L}_{\text{ELBO}}(\theta, \psi)$$
$$:= \mathbb{E}_{\mathcal{D}_{\text{tr}}}\left[\mathbb{E}_{q_\psi(\mathbf{z}|\mathbf{x})}\left[\log p_\theta(\mathbf{x}|\mathbf{z}) + \log\frac{p_\theta(\mathbf{z}|\mathbf{y}, e)}{q_\psi(\mathbf{z}|\mathbf{x})}\right]\right]. \quad (10)$$

Concerning the posterior, we assume a multi-normal distribution as suggested in (Khemakhem et al., 2020a; Kong et al., 2022) and take Multilayer perceptron (MLP) to estimate the mean and covariance matrix. The decoder of VAE, also parameterized by an MLP, maps the re-parameterized latent codes to the observed space as follows.

$$\mathbf{z} \sim q_\psi(\mathbf{z}|\mathbf{x}) := \mathcal{N}(\mu_\psi(\mathbf{x}), \Sigma_\psi(\mathbf{x})), \ \mathbf{x} = g(\mathbf{z}). \quad (11)$$

The first term of Eq. 10 maximizes the reconstruction performance, measured by the probability of $\mathbf{x}$ on $P(\mathbf{x})$. In contrast, the second term tries to minimize the distance between the posterior and the prior of $\mathbf{z}$. From Eqs. (3,4), we have $p(\mathbf{z}_c|\mathbf{y}) = \frac{p(\hat{\mathbf{z}}_c)}{|\det(\mathbf{J}(f_\mathbf{y}^*))|}$ and $p(\mathbf{z}_s|\mathbf{y}, e) = \frac{p(\hat{\mathbf{z}}_{s_i})}{|\det(\mathbf{J}(f_{(\mathbf{y}, e)}^\#))|}$ for each $\mathbf{z}_{s_i}$, which means that we can transform the probability density of $\mathbf{z}$ to $p(\hat{\mathbf{z}})$. To achieve this, we estimate **each function** in $f_\mathbf{y}^*$ and $f_{(\mathbf{y}, e)}^\#$ by a **distinct** flow-based architecture (Durkan et al., 2019), which ensures that the inverse $f_\mathbf{y}^{*-1}$ and $f_{(\mathbf{y}, e)}^{\#}{}^{-1}$ can be easily calculated. Then, we have

$$\hat{\mathbf{z}}_c = f_\mathbf{y}^{*-1}(\mathbf{z}_c), \quad \hat{\mathbf{z}}_s = f_{(\mathbf{y}, e)}^{\#}{}^{-1}(\mathbf{z}_s), \quad (12)$$

Then, we can compute the posterior $q_\psi(\hat{\mathbf{z}}_s|\mathbf{x})$, $q_\psi(\hat{\mathbf{z}}_c|\mathbf{x})$ by performing Eq. 11 and Eq. 12. We assume both the prior $p(\hat{\mathbf{z}}_c)$ and $p(\hat{\mathbf{z}}_s)$ as standard normal distributions denoted as $\mathcal{N}(\mathbf{0}_{n_c}, \mathbf{I}_{n_c})$ $\mathcal{N}(\mathbf{0}_{n_s}, \mathbf{I}_{n_s})$, respectively. Overall, we reformulate the negative ELBO, re-named as VAE loss, by

$$\mathcal{L}_{\text{VAE}} = \mathbb{E}_{\mathcal{D}_{\text{tr}}}\left[\mathbb{E}_{\mathbf{z}\sim q_\psi}\log p(\mathbf{x}|\mathbf{z}) + \beta \cdot \text{KL}\left(q_{\psi, f_\mathbf{y}^* f_{(\mathbf{y}, e)}^\#}(\hat{\mathbf{z}}|\mathbf{x}) \| p(\hat{\mathbf{z}})\right)\right]. \quad (13)$$

The value of $\beta$, which is a hyper-parameter, is used in the computation of KL divergence between the posterior and prior of $\hat{\mathbf{z}}$. This computation is made tractable by utilizing the flow model. In this case, we utilize the posterior of $\hat{\mathbf{z}}$ instead of $\mathbf{z}$ to avoid making any assumptions about the prior of $\mathbf{z}_c$.

To estimate the invariant predictor $\phi$, we also employ an MLP that takes the content factors $\tilde{\mathbf{z}}_c$ as input and predicts the corresponding label $\mathbf{y}$. The Cross-entropy loss $\mathcal{L}_{\text{cls}}$ is used for this purpose. Overall, the learning objective of our model would be

$$\mathcal{L}_{\text{overall}} = \mathcal{L}_{\text{cls}} + \alpha \cdot \mathcal{L}_{\text{VAE}}, \tag{14}$$

where $\alpha$ is a hyper-parameter that controls the balance between the loss terms.

## 6 EXPERIMENTS

Using synthesized data, we first verify the block identifiability of the content factors and the linear identifiability of style factors. Following (Sun et al., 2021), we also test our proposed method on a synthetic dataset named Colored MNIST (C-MNIST). Then, we evaluate the performance of our method on real-world datasets, including PACS and Office-Home, to demonstrate its ability to generalize to new domains with distributional shifts. Appendix E provides more details.

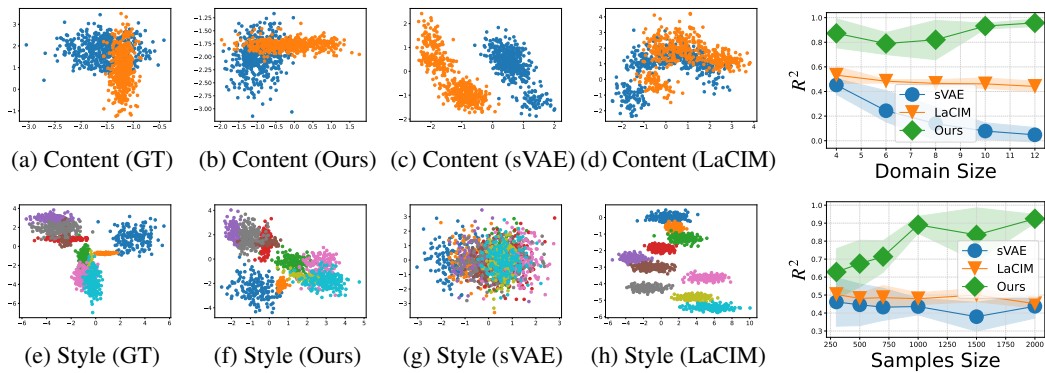

(a) Content (GT)   (b) Content (Ours)   (c) Content (sVAE)   (d) Content (LaCIM)

(e) Style (GT)   (f) Style (Ours)   (g) Style (sVAE)   (h) Style (LaCIM)

Figure 3: Identify the latent variables, including the content and style variables. The ground-truth distributions of the content and style variables are depicted in (a) and (e), respectively. Different colors mean different classes for the content variables. For the style variables, the different colors mean the different combinations of $(\mathbf{y}, e)$.

Figure 4: Identifiability results of the content variables among three methods on the synthetic data.

### 6.1 SYNTHETIC DATA

**Data Generation.** We generate the synthetic data following the data generation procedure outlined in Figure 1 with latent variable sizes set to $n_c = n_s = 2$. The style variable, from a general exponential family, is modulated by adjusting the mean and variance of a multi-normal distribution. We follow the same TCL data generation process in (Khemakhem et al., 2020a). That is $\mathbf{z}_s \sim \mathcal{N}(\mu_{(\mathbf{y},e)}, \sigma^2_{(\mathbf{y},e)}\mathbf{I})$, wherein we obtain $\mu_{(\mathbf{y},e)} \sim \text{Uniform}(-3, 3)$ and $\sigma^2_{(\mathbf{y},e)} \sim \text{Uniform}(0.1, 3)$ for each combination of $(\mathbf{y}, e)$. For simplicity, we generate $\mathbf{z}_c$ similarly, with mean and variance depending on the variation of $\mathbf{y}$. The mixing function $g$ is implemented using a 2-layer MLP with Leaky-ReLU activation, similar to (Hyvarinen & Morioka, 2016; Khemakhem et al., 2020a).

**Evaluation metrics.** We introduce two evaluation measures: the coefficient of determination ($R^2$) and the Mean Correlation Coefficient (MCC). MCC assesses the quality of component-wise identifiability of $\mathbf{z}_s$ by matching each learned component with the corresponding source component through cost optimization under permutation. To evaluate the block-wise identifiability of $\mathbf{z}_c$, we use kernel ridge regression to predict the true $\mathbf{z}_c$ and $\mathbf{z}_s$ from the estimated $\mathbf{z}_c$ and obtain the $R^2$ coefficient of determination. A higher value for both measures indicates better recovery of the latent variables. We repeat each experiment over 5 random seeds.

**Results.** In Figure 3, we set the total number of classes as $n_{\mathbf{y}} = 2$, the total number of domains as $m = 5$, and generate 2000 observations for each class. We visualize the distribution of the true latent variables and the recovered ones. Note that block-identifiability and linear identifiability do not guarantee perfect identification of the latent variables but rather the ability to identify them within certain block-wise or component-wise non-linear transformations. As shown in Figure 3, our method can recover the ground-truth latent factors up to trivial indeterminacies (rotation and sign flip). LaCIM (Sun et al., 2021) is unable to distinguish between different classes of content factors since the violation of the informative label distribution. On the other hand, the (supervised) sVAE approach attempts to extract distinct information among different classes, but these features may not remain consistent across various domains. Additionally, as shown in Figure 4, and Figure 5, the recovery quality ($R^2$, MCC) of $\mathbf{z}_c$ and $\mathbf{z}_s$ improve as the number of domains or observations increases, supporting our hypothesis that a sufficient combination of $(\mathbf{y}, e)$ is needed to identify the style factors and further enhance the recovery and isolation of the content factors.

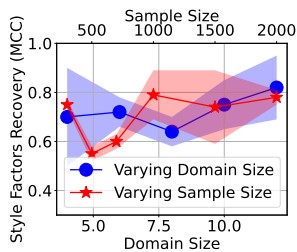

Figure 5: The recovery of style factors w.r.t varying domain size and sample size.

**Failure cases.** Our method's identifiability is mainly influenced by the sample size and sufficient variability, as per Theorem 1. When we set the domain size as $m = 2$ to intentionally violate the sufficient variability assumption, we observed $R^2$ values of $0.65$ for the recovered content factors and $0.58$ for the MCC of the recovered style factors. Similarly, when we reduced the sample size of each class to only $50$, the $R^2$ values of the recovered content factors and MCC of the recovered style factors were $0.56$ and $0.44$ respectively. These experiments highlight the critical role that our assumptions play in the successful recovery and separation of the latent variables.

## 6.2 COLORED MNIST

**Data Generation.** We construct the Colored `MNIST` dataset based on `MNIST` by relabeling digits $0 - 4$ as $\mathbf{y} = 0$ and $5 - 9$ as $\mathbf{y} = 1$ and then color a certain percentage of images with $\mathbf{y} = 0$ as green and the rest as red, and repeat this process for images with $\mathbf{y} = 1$. In the training datasets, we set $m = 2$, and set $p^{e_1} = 0.95$ and $p^{e_1} = 0.99$ for the two domains respectively. For the testing domain, we set $p^e = 0.1$. This dataset aligns with our data generation model, where label and domain influence the style variable (colorization of digits), and the label only controls the class.

Table 1: Test accuracy (%) on Colored `MNIST`.

| Algorithm | ACC | Params |
|---|---|---|
| ERM (Vapnik, 1991) | $91.9 \pm 0.9$ | 1.12M |
| DANN (Ganin et al., 2016) | $84.8 \pm 0.7$ | 1.1M |
| MMD-AAE (Li et al., 2018b) | $92.5 \pm 0.8$ | 1.23M |
| DIVA (Ilse et al., 2020) | $86.1 \pm 1.0$ | 1.69M |
| IRM (Arjovsky et al., 2019) | $92.9 \pm 1.2$ | 1.12M |
| sVAE (Sun et al., 2021) | $93.6 \pm 0.9$ | 0.92M |
| LaCIM (Sun et al., 2021) | $96.6 \pm 0.3$ | 0.92M |
| Ours | $\mathbf{97.2 \pm 0.3}$ | **0.73M** |

**Baselines and Results.** We evaluate our method against several state-of-the-art methods such as ERM (Vapnik, 1991), domain-adversarial neural network (DANN) (Ganin et al., 2016), Maximum Mean Discrepancy with Auto-Encoder (MMD-AAE) (Li et al., 2018b), Domain invariant Variational Autoencoders (DIVA) (Ilse et al., 2020), Invariant Risk Minimization (IRM) (Arjovsky et al., 2019), Supervised VAE (sVAE) and Latent Causal Invariance Models (LaCIM) (Sun et al., 2021). The results of these baseline methods are taken from (Sun et al., 2021). As shown in Table 1, our method, which has fewer parameters, achieves superior results compared to all other baselines, indicating that our model and data match well.

## 6.3 REAL-WORLD DATA

**Datasets, Baselines, and Setup.** We conduct experiments on two image classification datasets, `PACS` and `Office-Home`. These datasets consist of multiple domains with distinct styles and many classes. To ensure fair comparisons, we utilize the popular model selection method, *training-domain validation set*, where one domain is designated as the test domain, and data from all other domains are used for training the model. To ensure consistency, we use the ResNet50 (He et al., 2016) model, pretrained on `ImageNet`, as the backbone model for all methods, fine-tuning it for different tasks.

**Results.** The results of all methods on the PACS and Office-Home datasets are presented in Table 2 and Table 3. Additional results can be found in Table 4. Empirically, our method demonstrates

Table 2: Test accuracy(std) (%) on `PACS`.

| Algorithm | A | C | P | S | Avg. |
|---|---|---|---|---|---|
| ERM (Vapnik, 1991) | 84.7(0.4) | 80.8(0.6) | 97.2(0.3) | 79.3(1.0) | 85.5 |
| IRM (Arjovsky et al., 2019) | 84.8(1.3) | 76.4(1.1) | 96.7(0.6) | 76.1(1.0) | 83.5 |
| MMD (Li et al., 2018b) | 86.1(1.4) | 79.4(0.9) | 96.6(0.2) | 76.5(0.5) | 84.6 |
| DANN (Ganin et al., 2016) | 86.4(0.8) | 77.4(0.8) | 97.3(0.4) | 73.5(2.3) | 83.6 |
| CORAL (Sun et al., 2016) | 88.3(0.2) | 80.0(0.5) | 97.5(0.3) | 78.8(1.3) | 86.2 |
| SagNet (Nam et al., 2021) | 87.4(1.0) | 80.7(0.6) | 97.1(0.1) | 80.0(0.4) | 86.3 |
| RSC (Huang et al., 2020) | 85.4(0.8) | 79.7(1.8) | 97.6(0.3) | 78.2(1.2) | 85.2 |
| EQRM (Eastwood et al., 2022) | 86.5(0.4) | 82.1(0.7) | 96.6(0.2) | 80.8(0.2) | 86.5 |
| CB (Wang et al., 2023) | 87.8(0.8) | 81.0(0.1) | 97.1(0.4) | 81.1(0.8) | 86.7 |
| MADG (Dayal et al., 2023) | 87.8(0.5) | 82.2(0.6) | 97.7(0.3) | 78.3(0.4) | 86.5 |
| Ours | 88.7(0.3) | 80.6(0.8) | 97.7(0.4) | 82.6(0.4) | 87.4 |

Table 3: Test accuracy(std) (%) on `OfficeHome`.

| Algorithm | A | C | P | R | Avg. |
|---|---|---|---|---|---|
| ERM (Vapnik, 1991) | 61.3(0.7) | 52.4(0.3) | 75.8(0.1) | 76.6(0.3) | 66.5 |
| IRM (Arjovsky et al., 2019) | 58.9(2.3) | 52.2(1.6) | 72.1(0.1) | 74.0(0.3) | 64.3 |
| MMD (Li et al., 2018b) | 60.4(0.2) | 53.3(0.3) | 74.3(0.1) | 77.4(0.6) | 66.3 |
| DANN (Ganin et al., 2016) | 59.9(1.3) | 53.0(0.3) | 73.6(0.7) | 76.9(0.5) | 65.9 |
| CORAL (Sun et al., 2016) | 65.3(0.4) | 54.4(0.5) | 76.5(0.1) | 78.4(0.5) | 68.7 |
| SagNet (Nam et al., 2021) | 63.4(0.2) | 54.8(0.4) | 75.8(0.4) | 78.3(0.3) | 68.1 |
| RSC (Huang et al., 2020) | 60.7(1.4) | 51.4(0.3) | 74.8(1.1) | 75.1(1.3) | 65.5 |
| EQRM (Eastwood et al., 2022) | 60.5(0.1) | 56.0(0.2) | 76.1(0.4) | 77.4(0.3) | 67.5 |
| CB (Wang et al., 2023) | 65.6(0.6) | 56.5(0.6) | 77.6(0.3) | 78.8(0.7) | 69.6 |
| MADG (Dayal et al., 2023) | 67.6(0.2) | 54.1(0.6) | 78.4(0.3) | 80.3(0.5) | 70.1 |
| Ours | 64.8(0.7) | 56.1(0.3) | 78.4(0.6) | 79.8(0.3) | 69.8 |

the highest average performance compared to all other baseline methods. Comparisons with more baselines are included in the Appendix D.

**Parameter Sensitivity.** Figure 6 illustrates the sensitivity of our method to hyperparameters $\alpha$ and $\beta$. Specifically, we vary $\alpha$ over the range $1e-5, 1e-4, 1e-3, 1e-3$ and $\beta$ over the range $0.1, 0.5, 1, 5$. It can be observed that our method achieves competitive performance robustly across a wide range of hyper-parameter values. Figure 7 presents results on various style dimensions. The performance of the model is seen to degrade at large and small values, aligning with the proposed sufficient variability assumption and minimal change assumption outlined in previous work (Kong et al., 2022; Xie et al., 2023).

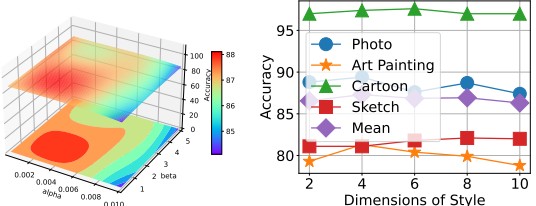

Figure 6: The sensitivity of our method to hyperparameters $\alpha$ and $\beta$.

Figure 7: The impact of the changing style dimension during learning.

# 7 RELATED WORKS

There are three mainstreams in recent DG research, including (1) data augmentation (Yue et al., 2019) to increase the training dataset size artificially; (2) invariant feature disentanglement or extraction (Arjovsky et al., 2019) to learn features invariant across domains; and (3) Advanced learning strategies such as meta-learning (Li et al., 2018a), ensemble learning (Zhou et al., 2021), and self-supervised learning (Bucci et al., 2021). Our method is a part of the domain generalization techniques that strive to find a model with invariant features across domains. To this end, DICA (Muandet et al., 2013) first proposed to learn an invariant transformation of the data across domains, effectively generalizing it to new domains in theory and practice. Subsequently, other methods have been proposed, such as maximum mean discrepancy (Li et al., 2018b) and adversarial learning (Li et al., 2018c). Invariant Risk Minimization (Arjovsky et al., 2019) is based on invariant causal prediction (Peters et al., 2016) and aims to learn an optimal classifier that remains invariant across all domains in the representation space. More related work can be seen in Appendix A.

# 8 DISCUSSION

**Limitations:** Our model has two potential limitations. Firstly, we only allow dependence between the content and style factors through the label effect, which may not be sufficient to capture all relevant dependence in real-world scenarios. Secondly, the model's identifiability is dependent on the number of environments and the label and also the infinite observation assumptions, which could limit its practical applicability if the number of style factors is quite large.

**Conclusion:** This paper presents a novel approach for DG by first introducing a novel data generation model, where the latent space is partitioned into invariant content and variant style components across domains. The proposed method uses a VAE framework to estimate the latent factors and can extract the invariant content components for more stable prediction effectively. Theoretical results and numerical simulations demonstrate the effectiveness of the proposed approach, which is further supported by experimental results on real-world datasets, showing its potential for developing models that exhibit a higher degree of generality.

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

# Appendix

## Table of Contents

## A  MORE RELATED WORKS

### A.1  FEATURE DISENTANGLEMENT.

DIVA (Ilse et al., 2020) separates the feature space into the domain, category, and other features and then disentangles them using a VAE framework. Subsequent studies, such as (Nam et al., 2021; Zhang et al., 2022), have attempted to separate semantic and stylistic features in the latent space using a generative model. However, these methods still lack theoretical guarantees to recover the semantic features. In the context of self-supervised learning, Von Kügelgen et al. (2021) studied the identification of the unchanging, shared portion of latent variables, known as the identifiability of the content part, in a block-wise manner. Later, Kong et al. (2022) extended this work for domain adaptation by partially identifying latent variables based on the property of minimal changes in causal mechanisms. LaCIM (Sun et al., 2021) assumes that content variables can only be identified from the label distribution, which is relatively strong, to achieve latent variable separation. iCaRL (Lu et al., 2021) assumes the invariance of the underlying causal diagram for prediction while allowing the content variables to change across environments.

### A.2  NONLINEAR ICA.

Nonlinear ICA is a technique used to uncover independent latent variables from data generated through nonlinear transformations of underlying independent variables. However, the general problem is ill-posed and cannot be uniquely solved without additional assumptions (Hyvärinen & Pajunen, 1999). Recent research has employed various methods to overcome this, such as utilizing additional observable variables such as time index (Hyvarinen & Morioka, 2016; 2017), auxiliary label (Hyvarinen et al., 2019; Khemakhem et al., 2020a;b), and multi-view information (Gresele et al., 2020). Additionally, some studies have restricted the mixing function to identify latent sources (Zheng et al.,

2022). The finite-sample identifiability of the nonlinear ICA model has been analyzed in (Lyu & Fu, 2022), and the unknown intrinsic problem dimension has been studied in (Sorrenson et al., 2020).

### A.3 CAUSALITY IN DG.

Peters et al. (2016) employs causality knowledge to model the distribution shifts across different domains through interventions. Including most of the following works (Arjovsky et al., 2019; Magliacane et al., 2018; Bühlmann, 2020), they assume a shared structural causal model (SCM) underlying all domains, where each domain corresponds to a specific intervention over some variables. Achieving domain generalization (DG) relies on the assumption that the causal mechanism of the label remains unchanged while allowing interventions on other observed variables. Both Sun et al. (2021) and Lu et al. (2021) expand this concept of invariant features to the latent space, which aligns well with sensory-level data like images and audio.

## B COLORED MNIST UNDER OUR MODEL

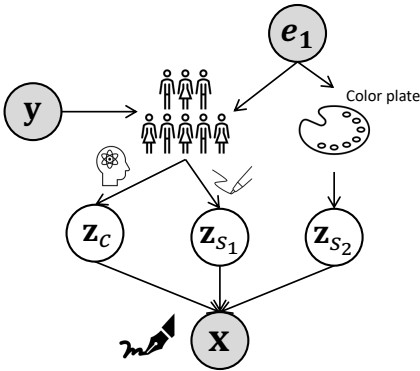
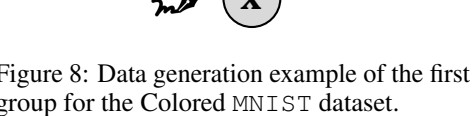
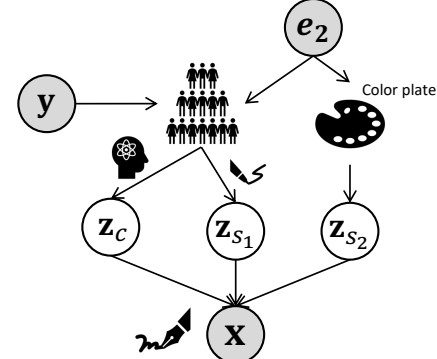

Figure 8: Data generation example of the first group for the Colored MNIST dataset.

Figure 9: Data generation example of the second group for the Colored MNIST dataset.

**Example 1** *Consider two groups constructing the overall dataset, wherein the two sub-datasets (domains) are denoted as $\mathbb{D}^{e_1}$ and $\mathbb{D}^{e_2}$, respectively. People write down each corresponding digit $\mathbf{x}$ according to the instruction (label) $\mathbf{y}$, which means the content factor $\mathbf{z}_c$ (shape) of the digits $\mathbf{x}$ only depends on $\mathbf{y}$. Moreover, different people have different writing styles $\mathbf{z}_{s_1}$, such as the inclination and the boldness of the digit, and tend to use different colors of pens, denoted as $\mathbf{z}_{s_2}$, when writing different digits (means different $\mathbf{y}$). Therefore, $\mathbf{z}_{s_2}$ would only be influenced by $e$ while $\mathbf{z}_{s_1}$ could be affected by both $e$ and $\mathbf{y}$. Then, given the shape $\mathbf{z}_c$, writing style $\mathbf{z}_{s_1}$ and color $\mathbf{z}_{s_2}$ factors, the corresponding observation $\mathbf{x}$ can be obtained.*

We visualize the data generation process introduced in Example B in Figure 8 and Figure 9. Then, we show some observations sampled from the two different training sets and the test set in Figure 10, Figure 11 and Figure 12. Different from the modeling method proposed in (Sun et al., 2021; Kong et al., 2022), we propose an anti-causal prediction (Schölkopf et al., 2012) modeling to depict the data generation process.

## C PROOF OF THE IDENTIFIABILITY.

In this section, we mainly give the detailed proof of the Theorem 1.

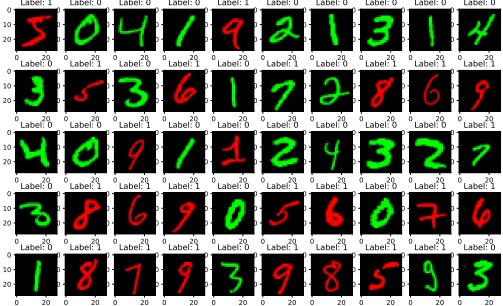

Figure 10: Some observations sampled from the first training domain.

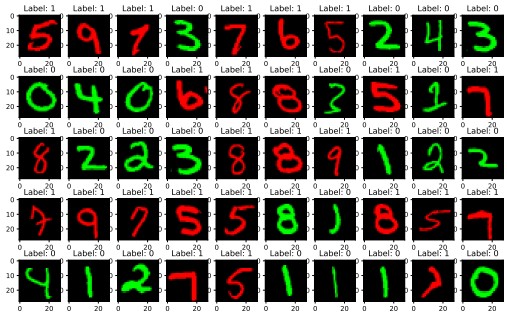

Figure 11: Some observations sampled from the second training domain.

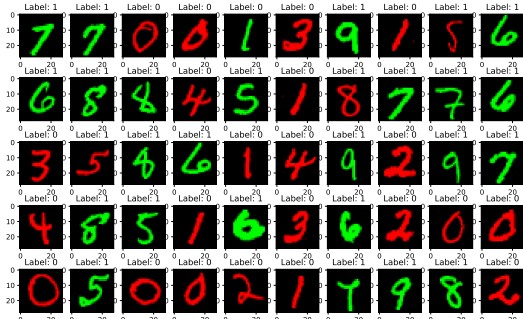

Figure 12: Some observations sampled from the testing domain.

## C.1 PRELIMINARIES

Recall that we assume the distribution of $\mathbf{z}_s$ has a density of the form

$$p_{\mathbf{T},\boldsymbol{\lambda}}(\mathbf{z}_s|\mathbf{y},e) = \prod_{i=1}^{n_s} \frac{Q_i(\mathbf{z}_{s_i})}{\Gamma_i(\mathbf{y},e)} \exp\left[\sum_{j=1}^{k} T_{ij}(\mathbf{z}_{s_i})\lambda_{ij}(\mathbf{y},e)\right],$$

Furthermore, we suppose that the probability density function of the style factors $\mathbf{z}_s$ given the label and domain variable, $p(\mathbf{z}_s|\mathbf{y},e)$ belongs to the strongly exponential families defined as follows

**Definition 3 (Strongly exponential)** *(Khemakhem et al., 2020a, Definition 4) We say that an exponential family distribution is* strongly exponential *if for any subset $\mathcal{X}$ of $\mathbb{R}$ the following is true:*

$$\left(\exists\, \boldsymbol{\theta} \in \mathbb{R}^k \mid \forall x \in \mathcal{X}, \langle \mathbf{T}(x), \boldsymbol{\theta} \rangle = const\right) \implies (\Lambda(\mathcal{X}) = 0 \text{ or } \boldsymbol{\theta} = 0) \qquad (15)$$

*where $\Lambda$ is the Lebesgue measure.*

Also, note that if part of the $\mathbf{z}_s$ are modulated by the auxiliary variable $(\mathbf{y},e)$, then we can re-write the density form as

$$p(\mathbf{z}_s|\mathbf{y},e) = \prod_{i=1}^{n_s} \frac{Q_i(\mathbf{z}_{s_i})}{\Gamma_i(\mathbf{y},e)} \exp\left[\sum_{i=1}^{n_s^{\star}} \sum_{j=1}^{k} T_{ij}(z_{s_i})\lambda_{ij}(\mathbf{y},e)\right]. \qquad (16)$$

Actually, the term $\exp\left[\sum_{i=n_s^{\star}}^{n_s} \sum_{j=1}^{k} T_{ij}(z_{s_i})\lambda_{ij}(\mathbf{y},e)\right]$ can be absorbed into the first term. That is to say, this expression is useful for dimension reduction. While in the following, we still use $n_s$ instead of $n_s^{\star}$ for simplicity. Moreover, we rely on the following Lemmas from (Khemakhem et al., 2020a;b), which we list below for the completeness of our proof.

**Lemma 1** *Consider an exponential family distribution with $k \geq 2$ components. Then the components of the sufficient statistic $\mathbf{T}$ are linearly independent.*

**Lemma 2** *Consider a* strongly exponential *family distribution such that its sufficient statistic* $\mathbf{T}$ *is differentiable almost surely. Then* $T_i' \neq 0$ *almost everywhere on* $\mathbb{R}$ *for all* $1 \leq i \leq k$.

**Lemma 3** *Consider a strongly exponential distribution of size* $k \geq 2$ *with sufficient statistic* $\mathbf{T}(x) = (T_1(x), \ldots, T_k(x))$. *Further, assume that* $\mathbf{T}$ *is* differentiable *almost everywhere. Then there exist* $k$ *distinct values* $x_1$ *to* $x_k$ *such that* $(\mathbf{T}'(x_1), \ldots, \mathbf{T}'(x_k))$ *are linearly independent in* $\mathbb{R}^k$.

**Lemma 4** *Consider a strongly exponential distribution of size* $k \geq 2$ *with sufficient statistic* $\mathbf{T}$. *Further assume that* $\mathbf{T}$ *is* twice differentiable *almost everywhere. Then*

$$\dim\left(\mathrm{span}\left(\left(T_i'(x), T_i''(x)\right)^T, 1 \leq i \leq k\right)\right) \geq 2 \tag{17}$$

*almost everywhere on* $\mathbb{R}$.

**Lemma 5** *Consider* $n$ *strongly exponential distributions of size* $k \geq 2$ *with respective sufficient statistics* $\mathbf{T}_j = (T_{j,1}, \ldots T_{j,k})$, $1 \leq j \leq n$. *Further consider that the sufficient statistics are* twice differentiable. *Define the vectors* $\mathbf{e}^{(j,i)} \in \mathbb{R}^{2n}$, *such that* $\mathbf{e}^{(j,i)} = \left(0, \ldots, 0, T_{j,i}', T_{j,i}'', 0, \ldots, 0\right)$, *where the non-zero entries are at indices* $(2j, 2j+1)$. *Let* $\mathbf{x} := (x_1, \ldots, x_n) \in \mathbb{R}^n$. *Then the matrix* $\bar{\mathbf{e}}(\mathbf{x}) := (\mathbf{e}^{(1,1)}(x_1), \ldots, \mathbf{e}^{(1,k)}(x_1), \ldots \mathbf{e}^{(n,1)}(x_n), \ldots, \mathbf{e}^{(n,k)}(x_n))$ *of size* $(2n \times nk)$ *has rank* $2n$ *almost everywhere on* $\mathbb{R}^n$.

### C.2 Proof of the Linear Identifiability of the style part.

We first provide proof of the linear identifiability of the style variable $\mathbf{z}_s$ since separating and recovering $\mathbf{z}_c$ relies on this result. The following three steps complete the proof.

**[Step 1.]** We follow the same way as in (Khemakhem et al., 2020a) to define $\mathrm{vol}(A) = \sqrt{\det A^T A}$, and when $A$ is invertible, $\mathrm{vol}\, A = |\det A|$. The matrix volume can be used in the change of variable formula to replace the absolute determinant of the Jacobian. This is most useful when the Jacobian is a rectangular matrix ($n_s + n_c < d$). Suppose we have two sets of parameters $(g, \theta_c, \theta_s := (\mathbf{T}, \boldsymbol{\lambda}))$ and $(\tilde{g}, \tilde{\theta}_c, \tilde{\theta}_s := (\tilde{\mathbf{T}}, \tilde{\boldsymbol{\lambda}}))$ such that $p_{g,\theta_c,\theta_s}(\mathbf{x}|\mathbf{y}, e) = p_{\tilde{g},\tilde{\theta}_c,\tilde{\theta}_s}(\mathbf{x}|\mathbf{y}, e)$ for all pairs $(\mathbf{x}, \mathbf{y}, e) \in \mathcal{X} \times \mathcal{Y} \times \mathcal{E}_{\mathrm{tr}}$. Then we have

$$\int_{\mathcal{Z}} p_{\theta_c,\theta_s}(\mathbf{z}|\mathbf{y}, e) p_g(\mathbf{x}|\mathbf{z}) \mathrm{d}\mathbf{z} = \int_{\mathcal{Z}} p_{\tilde{\theta}_c,\tilde{\theta}_s}(\mathbf{z}|\mathbf{y}, e) p_{\tilde{g}}(\mathbf{x}|\mathbf{z}) \mathrm{d}\mathbf{z} \tag{18}$$

$$\implies \int_{\mathcal{Z}} p_{\theta_c,\theta_s}(\mathbf{z}|\mathbf{y}, e) p_\epsilon(\mathbf{x} - g(\mathbf{z})) \mathrm{d}\mathbf{z} = \int_{\mathcal{Z}} p_{\tilde{\theta}_c,\tilde{\theta}_s}(\mathbf{z}|\mathbf{y}, e) p_\epsilon(\mathbf{x} - \tilde{g}(\mathbf{z})) \mathrm{d}\mathbf{z} \tag{19}$$

$$\implies \int_{\mathcal{X}} p_{\theta_c,\theta_s}(g^{-1}(\bar{\mathbf{x}})|\mathbf{y}, e) \mathrm{vol}\, \mathbf{J}_{g^{-1}}(\bar{\mathbf{x}}) p_\epsilon(\mathbf{x} - \bar{\mathbf{x}}) \mathrm{d}\bar{\mathbf{x}} = \int_{\mathcal{X}} p_{\tilde{\theta}_c,\tilde{\theta}_s}(\tilde{g}^{-1}(\bar{\mathbf{x}})|\mathbf{y}, e) \mathrm{vol}\, \mathbf{J}_{\tilde{g}^{-1}}(\bar{\mathbf{x}}) p_\epsilon(\mathbf{x} - \bar{\mathbf{x}}) \mathrm{d}\bar{\mathbf{x}} \tag{20}$$

$$\implies \int_{\mathbb{R}^d} \tilde{p}_{\theta_c,\theta_s,g,\mathbf{y},e}(\bar{\mathbf{x}}) p_\epsilon(\mathbf{x} - \bar{\mathbf{x}}) \mathrm{d}\bar{\mathbf{x}} = \int_{\mathbb{R}^d} \tilde{p}_{\tilde{\theta}_c,\tilde{\theta}_s,\tilde{g},\mathbf{y},e}(\bar{\mathbf{x}}) p_\epsilon(\mathbf{x} - \bar{\mathbf{x}}) \mathrm{d}\bar{\mathbf{x}} \tag{21}$$

$$\implies (\tilde{p}_{\theta_c,\theta_s,g,\mathbf{y},e} * p_\epsilon)(\mathbf{x}) = (\tilde{p}_{\tilde{\theta}_c,\tilde{\theta}_s,\tilde{g},\mathbf{y},e} * p_\epsilon)(\mathbf{x}) \tag{22}$$

$$\implies F[\tilde{p}_{\theta_c,\theta_s,g,\mathbf{y},e}](\omega) \xi_\epsilon(\omega) = F[\tilde{p}_{\tilde{\theta}_c,\tilde{\theta}_s,\tilde{g},\mathbf{y},e}](\omega) \xi_\epsilon(\omega) \tag{23}$$

$$\implies F[\tilde{p}_{\theta_c,\theta_s,g,\mathbf{y},e}](\omega) = F[\tilde{p}_{\tilde{\theta}_c,\tilde{\theta}_s,\tilde{g},\mathbf{y},e}](\omega) \tag{24}$$

$$\implies \tilde{p}_{\theta_c,\theta_s,g,\mathbf{y},e}(\mathbf{x}) = \tilde{p}_{\tilde{\theta}_c,\tilde{\theta}_s,\tilde{g},\mathbf{y},e}(\mathbf{x}) \tag{25}$$

where:

- in Eq. equation 20, $\mathbf{J}$ denotes the Jacobian, and we made the change of variable $\bar{\mathbf{x}} = g(\mathbf{z})$ on the left-hand side, and $\bar{\mathbf{x}} = \tilde{g}(\mathbf{z})$ on the right-hand side.
- in Eq. equation 21, we introduced

$$\tilde{p}_{\theta_c,\theta_s,g,\mathbf{y},e}(\mathbf{x}) = p_{\theta_c,\theta_s}(g^{-1}(\mathbf{x})|\mathbf{y}, e) \mathrm{vol}\, \mathbf{J}_{g^{-1}}(\mathbf{x}) \mathbb{1}_{\mathcal{X}}(\mathbf{x}) \tag{26}$$

  on the left-hand side and similarly on the right-hand side.
- in Eq. equation 22, we used $*$ for the convolution operator.

- in Eq. equation 23, we used $F[.]$ to designate the Fourier transform, and where $\xi_\epsilon = F[p_\epsilon]$ (by definition of the characteristic function).
- in Eq. equation 24, we dropped $\xi_\epsilon(\omega)$ from both sides as it is non-zero almost everywhere (by assumption i).

Then, we have

$$p_{\theta_c,\theta_s}(g^{-1}|\mathbf{y},e)\mathrm{vol}(\mathbf{J}_{g^{-1}}(\mathbf{x})) = p_{\tilde{\theta}_c,\tilde{\theta}_s}(\tilde{g}^{-1}|\mathbf{y},e)\mathrm{vol}(\mathbf{J}_{\tilde{g}^{-1}}(\mathbf{x})) \tag{27}$$

**[Step 2.]** In Eq. 27, we take the logarithm on both sizes and replace $p_{\theta_c,\theta_s}$ by the formulation in Eq. 7 to have

$$\log \mathrm{vol}\,\mathbf{J}_{g^{-1}}(\mathbf{x}) + \log p_{\theta_c}(\mathbf{z}_c|\mathbf{y}) + \sum_{i=1}^{n_s}(\log Q_i(g_i(\mathbf{x})) - \log \Gamma_i(\mathbf{y},e) + \sum_{j=1}^{k} T_{ij}(g_i^{-1}(\mathbf{x}))\lambda_{ij}(\mathbf{y},e)) =$$

$$\log \mathrm{vol}\,\mathbf{J}_{\tilde{g}^{-1}}(\mathbf{x}) + \log p_{\tilde{\theta}_c}(\mathbf{z}_c|\mathbf{y}) + \sum_{i=1}^{n_s}(\log \tilde{Q}_i(\tilde{g}_i(\mathbf{x})) - \log \tilde{\Gamma}_i(\mathbf{y},e) + \sum_{j=1}^{k} \tilde{T}_{ij}(\tilde{g}_i^{-1}(\mathbf{x}))\tilde{\lambda}_{ij}(\mathbf{y},e)), \tag{28}$$

where $T_{ij}(g^{-1}(\mathbf{x}))$ only concerns the sufficient statistics of $\mathbf{z}_s$ and similar for $\tilde{T}_{ij}(\tilde{g}^{-1}(\mathbf{x}))$. Let $(\mathbf{y}^{(1)}, e^{(1)}), \cdots, (\mathbf{y}^{(1)}, e^{(n_{\mathbf{y}^{(1)}})}), \cdots, (\mathbf{y}^{(n_{\mathbf{y}})}, e^{(1)}), \cdots, (\mathbf{y}^{(n_{\mathbf{y}})}, e^{(n_{\mathbf{y}^{(n_{\mathbf{y}})}})})$ be the distinct points in the **sufficient variability assumption** (iv). Then for each give $\mathbf{y}^{(i)}$, we subtract the first equation for $(\mathbf{y}^{(i)}, e^{(1)})$ from the remaining $n_{\mathbf{y}^{(i)}} - 1$ equations to get

$\forall l \in \{2, \cdots, n_{y^{(1)}}\}$

$$\langle \mathbf{T}(g^{-1}(\mathbf{x})), \boldsymbol{\lambda}(\mathbf{y}^{(1)}, e^{(l)}) - \boldsymbol{\lambda}(\mathbf{y}^{(1)}, e^{(1)}) \rangle + \log \frac{\Gamma(\mathbf{y}^{(1)}, e^{(1)})}{\Gamma(\mathbf{y}^{(1)}, e^{(l)})} =$$

$$\langle \tilde{\mathbf{T}}(\tilde{g}^{-1}(\mathbf{x})), \tilde{\boldsymbol{\lambda}}(\mathbf{y}^{(1)}, e^{(l)}) - \tilde{\boldsymbol{\lambda}}(\mathbf{y}^{(1)}, e^{(1)}) \rangle + \log \frac{\tilde{\Gamma}(\mathbf{y}^{(1)}, e^{(1)})}{\tilde{\Gamma}(\mathbf{y}^{(1)}, e^{(l)})}$$

$\vdots$

$\forall l \in \{2, \cdots, n_{y^{(i)}}\}$

$$\langle \mathbf{T}(g^{-1}(\mathbf{x})), \boldsymbol{\lambda}(\mathbf{y}^{(i)}, e^{(l)}) - \boldsymbol{\lambda}(\mathbf{y}^{(i)}, e^{(1)}) \rangle + \log \frac{\Gamma(\mathbf{y}^{(i)}, e^{(1)})}{\Gamma(\mathbf{y}^{(i)}, e^{(l)})} =$$

$$\langle \tilde{\mathbf{T}}(\tilde{g}^{-1}(\mathbf{x})), \tilde{\boldsymbol{\lambda}}(\mathbf{y}^{(i)}, e^{(l)}) - \tilde{\boldsymbol{\lambda}}(\mathbf{y}^{(i)}, e^{(1)}) \rangle + \log \frac{\tilde{\Gamma}(\mathbf{y}^{(i)}, e^{(1)})}{\tilde{\Gamma}(\mathbf{y}^{(i)}, e^{(l)})}$$

$\vdots$

$\forall l \in \{2, \cdots, n_{y^{(n_{\mathbf{y}})}}\}$

$$\langle \mathbf{T}(g^{-1}(\mathbf{x})), \boldsymbol{\lambda}(\mathbf{y}^{(n_{\mathbf{y}})}, e^{(l)}) - \boldsymbol{\lambda}(\mathbf{y}^{(n_{\mathbf{y}})}, e^{(1)}) \rangle + \log \frac{\Gamma(\mathbf{y}^{(n_{\mathbf{y}})}, e^{(1)})}{\Gamma(\mathbf{y}^{(n_{\mathbf{y}})}, e^{(l)})} =$$

$$\langle \tilde{\mathbf{T}}(\tilde{g}^{-1}(\mathbf{x})), \tilde{\boldsymbol{\lambda}}(\mathbf{y}^{(n_{\mathbf{y}})}, e^{(l)}) - \tilde{\boldsymbol{\lambda}}(\mathbf{y}^{(n_{\mathbf{y}})}, e^{(1)}) \rangle + \log \frac{\tilde{\Gamma}(\mathbf{y}^{(n_{\mathbf{y}})}, e^{(1)})}{\tilde{\Gamma}(\mathbf{y}^{(n_{\mathbf{y}})}, e^{(l)})}$$

Then, we concatenate all these equations together. Let $A$ bet the matrix defined in assumption iv, and $\tilde{A}$ similarly defined for $\tilde{\boldsymbol{\lambda}}$ ($\tilde{A}$ is not necessarily invertible). Define $b_{(\mathbf{y}^{(i)}, e^{(j)})} = \log \frac{\tilde{\Gamma}(\mathbf{y}^{(i)}, e^{(1)})\Gamma(\mathbf{y}^{(i)}, e^{(l)})}{\tilde{\Gamma}(\mathbf{y}^{(i)}, e^{(1)})\Gamma(\mathbf{y}^{(i)}, e^{(l)})}$ and $\mathbf{b}$ the vector of all $b_{(\mathbf{y}^{(i)}, e^{(j)})}$ for $i = 1, \cdots, n_{\mathbf{y}}$ and $j = 1, \cdots, n_{\mathbf{y}^{(i)}}$ for each $i$. Then we get:

$$A^T \mathbf{T}(g^{-1}(\mathbf{x})) = \tilde{A}^T \tilde{\mathbf{T}}(\tilde{g}^{-1}(\mathbf{x})) + \mathbf{b} \tag{29}$$

It is easy to get $A$ is of size $(\sum_{i=1}^{n_{\mathbf{y}}} n_{\mathbf{y}^{(i)}} - n_{\mathbf{y}}) \times n_s$. We multiply both sides of equation 29 by the transpose of the inverse of $A^T$ from the left to find:

$$\mathbf{T}(g^{-1}(\mathbf{x})) = B\tilde{\mathbf{T}}(\tilde{g}^{-1}(\mathbf{x})) + \mathbf{c}$$
$$\mathbf{T}(\mathbf{z}) = B\tilde{\mathbf{T}}(\tilde{g}^{-1}(\mathbf{x})) + \mathbf{c} \tag{30}$$

where $B = A^{-T}\tilde{A}$ and $\mathbf{c} = A^{-T}\mathbf{b}$.

[**Step 3.**] Then, to complete the proof, we need to prove that $B$ is invertible by taking the gradient of Eq. (30) w.r.t. $\mathbf{z}$. And the Jacobian $\mathbf{J_T}$ of $\mathbf{T}$ is a matrix of size $kn_s \times d$, where $d$ is the dimension of $\mathbf{x}$. The column is independent as each $\mathbf{T}_i$ is a function of $z_{s_i}$, and thus the non-zero entries of each column are in different rows. That is to say, $\mathbf{J_T}$ has rank $d$. This is not enough to prove that $B$ is invertible. For that, we consider the functions $\mathbf{T}_i$ for which $k > 1$: for each of these functions, using Lemma 3, there exists points $z_{s_i}^{(1)}, \ldots, z_{s_i}^{(k)}$ such that $(\mathbf{T}'_i(z_{s_i}^{(1)}), \ldots, \mathbf{T}'_i(z_{s_i}^{(k)}))$ are independent. Collate these point into $k$ vectors $\mathbf{z}^{(j)} := (z_1^{(j)}, \ldots z_d^{(j)})$. We plug these vectors into equation equation 30 after differentiating it, and collate the $n_s k$ equations in vector form:

$$\mathbf{M} = B\tilde{\mathbf{M}} \tag{31}$$

where $\mathbf{M} := (\ldots, \mathbf{J_T}(\mathbf{z}^{(j)}), \ldots)$ and $\tilde{\mathbf{M}} := (\ldots, \mathbf{J}_{\tilde{g}^{-1}\circ g}(\mathbf{z}^{(j)}), \ldots)$. Now the matrix $\mathbf{M}$ is of size $k \times n_s k$, and it has exactly $k$ independent columns by definition of the points $\mathbf{z}^{(j)}$. This means that $\mathbf{M}$ is of rank $n_s k$, which implies that $\text{rank}(B) \geq n_s k$. Since $B$ is a $n_s k \times n_s k$ matrix, we conclude that $B$ is invertible.

For the case that $k = 1$, this directly means that $B$ is invertible as $B$ is of size $n_s \times n_s$. Hence, and the invertibility of $B$ mean that $(\tilde{g}, \tilde{\mathbf{T}}, \tilde{\boldsymbol{\lambda}}) \sim (g, \mathbf{T}, \boldsymbol{\lambda})$. However, notice that the identifiability of $g$ cannot be obtained as $\mathbf{T}(g^{-1}(\mathbf{x}))$ only concerns the last $n_s$ components of $g^{-1}(\mathbf{x})$. $\qquad\square$

### C.3 PROOF OF THE BLOCK IDENTIFIABILITY OF THE CONTENT PART.

We start our proof from the matched marginal distribution to develop the relation between $\mathbf{z}_c$ and $\hat{\mathbf{z}}_c$.

$$p(\mathbf{x}|\mathbf{y}, e) = p(\tilde{\mathbf{x}}|\mathbf{y}, e) \Leftrightarrow p(\zeta(\mathbf{z})|\mathbf{y}, e) = p(\tilde{\zeta}(\tilde{\mathbf{z}})|\mathbf{y}, e) \Leftrightarrow p(\mathbf{z}|\mathbf{y}, e)|\mathbf{J}_{\zeta^{-1}}| = p(\zeta^{-1} \circ \tilde{\zeta}(\tilde{\mathbf{z}})|\mathbf{y}, e)|\mathbf{J}_{\zeta^{-1}}|, \tag{32}$$

where $\tilde{\zeta}^{-1}$ denotes the estimated invertible generating function and $h := \zeta^{-1} \circ \tilde{\zeta}$ is the transformation between the true latent variable and the estimated one. Then the Jacobian of $h$ can be represented as

$$\mathbf{J}_h = \left[\begin{array}{c|c} \mathbf{A} := \frac{\partial \mathbf{z}_c}{\partial \tilde{\mathbf{z}}_c} & \mathbf{B} := \frac{\partial \mathbf{z}_c}{\partial \tilde{\mathbf{z}}_s} \\ \hline \mathbf{C} := \frac{\partial \mathbf{z}_s}{\partial \tilde{\mathbf{z}}_c} & \mathbf{D} := \frac{\partial \mathbf{z}_s}{\partial \tilde{\mathbf{z}}_s} \end{array}\right], \tag{33}$$

From the above conclusion that $\mathbf{z}_s$ can be component-wise recovered means that $\mathbf{C}$ is a zero matrix and $\mathbf{D}$ must be of full rank. To prove that $\mathbf{z}_c$ can be block-wise identifiable, we recommend readers to refer to the proof in (Kong et al., 2022).

We also define a smooth and injective function $\bar{h} := \tilde{\zeta}^{-1} \circ \zeta$. Until step 3, it has been proved that $\bar{h}_c(\cdot)$ does not depend on the style factors $\mathbf{z}_s$, where $\bar{h}_c(\cdot)$ means extract the first $n_c$ variables from $\bar{h}(\cdot)$. Then, we define a Jacobian matrix as

$$\mathbf{J}_{\bar{h}} = \left[\begin{array}{c|c} \mathbf{A} := \frac{\partial \hat{\mathbf{z}}_c}{\partial \mathbf{z}_c} & \mathbf{B} := \frac{\partial \hat{\mathbf{z}}_c}{\partial \mathbf{z}_s} \\ \hline \mathbf{C} := \frac{\partial \hat{\mathbf{z}}_s}{\partial \mathbf{z}_c} & \mathbf{D} := \frac{\partial \hat{\mathbf{z}}_s}{\partial \mathbf{z}_s} \end{array}\right], \tag{34}$$

Then, from the above proof, we know that $\mathbf{z}_c$ does not depend on $\mathbf{z}_s$. It follows that $\mathbf{B} = \mathbf{0}$. we can see an inverse mapping between $\mathbf{z}_s$ and $\hat{\mathbf{z}}_s$. That is to say, we have $\mathbf{C} = \mathbf{0}$. Therefore, there exists an invertible function $\bar{h}'_c$ between the estimated and true content factors such that $\tilde{\mathbf{z}}_c = \bar{h}'_c(\mathbf{z}_c)$, which means that $\mathbf{z}_c$ can be block-wise identifiable by $\hat{\zeta}^{-1}$.

## D MORE RESULTS

Due to the page limit, we provide more results reported on DomainBed in this section to show the effectiveness of our proposed method.

## E IMPLEMENTATION DETAILS

### E.1 SIMULATION DATA.

In the simulation, our VAE framework's encoder and decoder are 6-layer MLPs with a hidden dimension of 32 and Leaky-ReLU activation functions with $\alpha = 0.2$. The codes are from Beta-

Table 4: Comparisons to more baselines.

| Algorithm | PACS | OfficeHome |
|---|---|---|
| GroupDRO (Sagawa et al., 2019) | $84.4 \pm 0.8$ | $66.0 \pm 0.7$ |
| Mixup (Yan et al., 2020) | $84.6 \pm 0.6$ | $68.1 \pm 0.3$ |
| MLDG (Li et al., 2018a) | $84.9 \pm 1.0$ | $66.8 \pm 0.6$ |
| CDANN (Li et al., 2018c) | $82.6 \pm 0.9$ | $65.8 \pm 1.3$ |
| MTL (Blanchard et al., 2011) | $84.6 \pm 0.5$ | $66.4 \pm 0.5$ |
| ARM (Zhang et al., 2021) | $85.1 \pm 0.4$ | $64.8 \pm 0.3$ |
| VREx | $84.9 \pm 0.6$ | $66.4 \pm 0.6$ |
| Fish | $85.5 \pm 0.3$ | $68.6 \pm 0.4$ |
| Fishr | $85.5 \pm 0.4$ | $67.8 \pm 0.1$ |
| AND-mask | $84.4 \pm 0.9$ | $65.6 \pm 0.4$ |
| SAND-mask | $84.6 \pm 0.9$ | $65.8 \pm 0.4$ |
| self-Reg | $85.6 \pm 0.4$ | $67.9 \pm 0.7$ |
| CausIRL(CORAL) | $85.8 \pm 0.1$ | $68.6 \pm 0.3$ |
| CausIRL(MMD) | $84.0 \pm 0.8$ | $65.7 \pm 0.6$ |
| Ours | $\mathbf{87.4 \pm 0.4}$ | $\mathbf{69.8 \pm 0.5}$ |

VAE (Higgins et al., 2017)[1]. For the flow architecture used for the recovery of the factor style $\mathbf{z}_s$, we follow a similar fashion in (Kong et al., 2022). The component-wise flow implementation is based on the spline flows (Durkan et al., 2019) with monotonic linear rational splines to modulate the change components[2]. We use 8 bins for each linear spline and set the bound to be 5. For the flow architecture used to recover the content factors, it is unnecessary to take a component-wise structure. Therefore, we take a masked autoregressive flow. We use Adam to train our VAE framework and flow models for 50 epochs. The $\beta$ parameter of the KL divergence loss is set to 0.1. The learning rate is 0.002, and the weight decay parameter is $1e^{-4}$. To determine the mean correlation coefficient (MCC), we first find the correlation coefficient between every pair of source and latent components. We then solve a linear sum assignment problem to match each latent component with the source component that correlates best with it, correcting any permutations in the latent space. A high MCC indicates successful identification of the true parameters and recovery of the true sources, excluding point-wise transformations. For the $R^2$, identifying the transformation function is regarded as a regression problem using Gaussian kernel ridge regression. The Gaussian kernel is universal, making this a nonparametric regression technique that can approximate any nonlinear function well with enough data.

### E.2 Colored MNIST.

For a fair comparison with LaCIM and supervised VAE (Sun et al., 2021), we first take an encoder part to map the image into a latent space. The encoder consists of four blocks in total, wherein each block is the sequential Convolutional Layer (CL), Batch Normalization (BN), a ReLU structure, and max-pooling with stride 2. For all layers, the corresponding numbers of hidden layers are $32 - 64 - 128 - 256$. Then, we take a Fully-Connected Layer (FCL)-BN-ReLU-FCL structure to approximate the posterior $p(\mu(\mathbf{z})|\mathbf{x})$ and $p(\Sigma(\mathbf{z})|\mathbf{x})$. The decoder consists of three modules named (1) upsampling with stride 2; (2) four blocks of Transpose-Convolution, BN, and ReLU; (3) CL-BN-ReLU-Sigmoid followed by cropping step in order to make the image with the same size as the input dimension. And the predictor is implemented as the structure of FCL-BN-ReLU-FC-BN-ReLU-FC. More details can be seen in the official Github Repository [3]. For the flow part, we also take the same structure as those used in the simulation and keep all the parameters the same.

For the inference procedure, we also follow the same style presented in LaCIM (Sun et al., 2021) and iCaRL (Lu et al., 2021), which fixes the learned generator $p(\mathbf{x}|\mathbf{z})$ and optimize the likelihood of $p(\mathbf{x}|\mathbf{z}_c, \mathbf{z}_s)$ over $\mathcal{Z}_c \times \mathcal{Z}_s$ to infer the best combination

$$\max_{\mathbf{z}_c, \mathbf{z}_s} \log p(\mathbf{x}|\mathbf{z}_c, \mathbf{z}_s) + \lambda_c \|\mathbf{z}_c\| + \lambda_s \|\mathbf{z}_s\|, \tag{35}$$

where $\lambda_c$ and $\lambda_s$ are hyperparameters set to control the scale of the learned latent factors. For optimization, we first sample some candidates from $\mathcal{N}(0, I)$ and select the optimal one in terms

---

[1]https://github.com/1Konny/Beta-VAE

[2]https://github.com/bayesiains/nsf

[3]https://github.com/wubotong/LaCIM

of Eq. (35). Notice that this method is totally different from previous works that directly obtain the encoded embedding of the input. However, this method does not need to leverage the new coming data to train the parameters of the neural nets. Therefore, it is still can be seen as a domain generalization method instead of a domain adaptation method. A more detailed discussion can be seen in the discussion phase of iCaRL [4].

### E.3    PACS, OFFICE-HOME.

As illustrated in the main text, we take ResNet50 pretrained on ImageNet (He et al., 2016) as the backbone network and get the network output as the representation of each image. As noted in (Kong et al., 2022), it is quite hard to train VAE on high-resolution images directly; we also take the extracted features as the input of our VAE. The encoder of our VAE is taken as 2-layer FCL, and an FCL-BN-ReLU-FCL structure as the decoder. For the classifier, we directly take an FCL. For recovering the high-level latent feature $\hat{z}_c$ and $\hat{z}_s$, we take flow architectures to achieve these. For the style features, the deep sigmoidal Flow model, which is extended to a component-wise version (Kong et al., 2022), is used to learn the transformation between the high-level and middle-level latent variables (More details can be found in the official codes of iMSDA[5].). For the content factors, we just take a masked-based flow to learn the transformation.

For the hyperparameter search, the search space of the learning rates is set to $[0.01, 0.05, 0.1]$. For the search space of $\alpha$, we set it to $[1e-5, 1e-4, 1e-3, 1e-2]$. And the search space of $\beta$ is set as $[0.1, 0.5, 1, 5]$. After the validation, we simply fixed $\alpha = 1e-5$, $lr = 1e-2$, and $\beta = 0.1$ across all our experiments. For each experiment, we run three different seeds to get the average result and the standard variance. Moreover, we use the common trick that applying different learning rates to the new trainable modules and the backbone network. In detail, we fix the learning rate of the backbone network as $0.1$ times the regular learning rate.

## F    DISCUSSIONS ON OUR METHOD

In this section, we provided more detailed discussions of assumptions made in this paper and the limitations of our method.

### F.1    DISCUSSIONS ON ASSUMPTIONS.

*The exponential family prior and domain variability assumptions.*

In the context of the domain generalization (DG) problem considered in our paper, identifying and isolating content factors from style factors within the latent space emerges as a key method, which contributes to the learning of a robust predictor capable of safely generalizing data from previously unseen distributions.

However, the pursuit of model identifiability, particularly with latent variables, poses a formidable and ongoing challenge. Traditional VAE approaches often hinge on the mild assumption that the data distribution of latent variables adheres to a normal distribution, making the problem unsolvable (Hyvärinen & Pajunen, 1999). In contrast, nonlinear ICA methods introduce a different paradigm by constraining the prior latent variables to the exponential family, where sufficient statistics are controlled by auxiliary information, such as time index or domain index. This is a trade-off between the model assumptions and model identifiability.

In alignment with the Nonlinear ICA techniques (Khemakhem et al., 2020a;b), our model also adopts a similar prior assumption on latent variables to ensure identifiability and leverage the label information and domain details to control the sufficient statistics of the prior distribution. Nevertheless, it is important to note that in scenarios with a large number of latent variables, our model may encounter challenges in providing a robust guarantee for identifying these variables. Moreover, if the style factors are not affected by the label of observations, our method may also fail to identify the latent variables.

*Infinite observations assumptions.*

---

[4] https://openreview.net/forum?id=-e4EXDWXnSn&noteId=i–Pn6lIA4D
[5] https://proceedings.mlr.press/v162/kong22a.html

Small sample sizes pose a huge challenge and prevent the Nonlinear ICA methods from achieving model identifiability without further assumptions. Conversely, our theoretical framework assumes infinite data and posits a universal function approximator, both of which may be unattainable in practical learning scenarios. In such real-world settings, the analysis of estimation errors becomes crucial, yielding an upper bound that is contingent on the complexity and smoothness of the demixing function, along with the sample size.

In practical terms, it is intuitive that learning a more intricate demixing necessitates a larger volume of data. Simultaneously, the lack of smoothness in the demixing function exacerbates the complexity of the estimation process (Hyvärinen et al., 2023). However, to our knowledge, there is only one paper (Lyu & Fu, 2022) that delves into the finite sample convergence of Nonlinear ICA methods. Notably, their focus remains confined to contrast-based learning methods—a facet that poses a non-trivial challenge when extending the findings to our methodology.

## F.2 DISCUSSIONS ON OUR MODEL.

*Connection with HVAE.*

Our approach aligns with a two-level HVAE framework but introduces additional constraints, distinguishing it from the traditional HVAE.

Similarities between the two methods include: (1) Latent space structure: Both the two-level HVAE and our method incorporate two layers of latent space, following a similar data generation structure: from the high level to the middle level, and then to the observations. (2) Learning procedures: The learning procedures of both methods exhibit similarities, involving reconstruction terms for observations and a prior matching term aimed at minimizing the distributions of high-level latent variables. (3) Distribution Assumptions on high-level latent variables: Both methods share the assumption of an independent normal distribution for high-level latent variables.

However, notable differences exist (1) Objective of tasks: Our method addresses the domain generalization problem, with the primary goal of isolating content factors from style factors in the latent space of observations, utilizing content factors for label prediction. In contrast, HVAE focuses more on learning data generation models to generate new observations. (2) Data generation process: We specify a specific data generation process from the high-level space to the middle-level space, where the mapping functions of content factors are influenced by the label while the mapping functions of style factors are affected by both the domain index and label. (3) Distribution assumption on the style variables in the middle level: We introduce an assumption regarding the data distributions of style latent variables in the middle level, a key for identifying these factors and facilitating the isolation of content factors.

*How does our method outperform previous methods?*

Indeed, many previous DG methods assume that labels and non-causal features are independent. Then, these methods try to separate the causal features (label-related) and use them for prediction. However, a significant drawback of these methods emerges: they may include some specific (causal) features whose distributions may change across domains yet are still related to labels, which will lead to instability when applied to unseen domains.

To deal with this case, we propose a novel approach incorporating a two-level latent space to accurately capture the generation of unstable features. Our theoretical framework establishes the identifiability of these latent features. Notably, our model surpasses prior methods by accommodating the interdependence between content and style factors, thereby relaxing the independence assumptions.

