# OpenReview forum: "Domain Generalization via Content Factors Isolation: A Two-level Latent Variable Modeling Approach"
_ICLR.cc/2024/Conference — Submitted to ICLR 2024_

### Official Review · Reviewer_Vftx · 2023-10-24

**Soundness:** 3 good
**Presentation:** 2 fair
**Contribution:** 2 fair
**Rating:** 6
**Confidence:** 2

**Summary:**

This paper proposes to learn a two-level, hierarchical latent space with each layer partitioned into the content and style group. The content group controls the data-invariant features, while the style group controls the data style factors. Such a model aims to address the difficulty in domain generalization that label-specific and domain-related features are not well distinguished.

**Strengths:**

1. The paper is in general well-written and well-presented with the illustration figures.
2. The idea is motivated to address the practical issue in an active research field.
3. The author conducts various experiment settings, including toy data synthesis and ablation studies.

**Weaknesses:**

1. I don't have much experience in this particular research field, so based on my understanding, the main purpose of the paper is to learn a well-defined and smooth latent space that can distinguish the domain features and style features; therefore, the model can perform well when the underlying distribution shift happens. The two-level latent space seems to be related to the hierarchical VAEs, where multi-layer latent variables are used to learn different levels of data features. So, how does such a two-level latent space compare or connect to the hierarchical VAEs?

2. I understand learning a separated latent space for different data features can be beneficial to learning a smoother model manifold. But how does this model improve the performance of DG? The author mentioned that "The key to achieving DG based on our model is recovering the distribution of content factors and isolating them from style factors." An additional and detailed explanation would be good.

**Questions:**

Please see the Weaknesses.

---

> ### Author Response · Authors · 2023-11-18
> **Response to reviewer Vftx (part 1/2)**
>
> We are grateful for your support and detailed comments! We provide responses to address your comments below.
>
> **[Q1. How does such a two-level latent space compare or connect to the hierarchical VAEs?]**
>
> We appreciate your suggestion regarding the connection between our method and the Hierarchical VAE (HVAE), noting that our approach aligns with a two-level HVAE framework but introduces additional constraints, distinguishing it from the traditional HVAE.
>
> Similarities between the two methods include:  (1) **Latent space structure:** Both the two-level HVAE and our method incorporate two layers of latent space, following a similar data generation structure: from the high-level to the middle level, and then to the observations. (2) **Learning procedures:** The learning procedures of both methods exhibit similarities, involving reconstruction terms for observations and a prior matching term aimed at minimizing the distributions of high-level latent variables. (3) **Distribution Assumptions on high-level latent variables:** Both methods share the assumption of an independent normal distribution for high-level latent variables.
>
> However, notable differences exist: (1) **Objective of tasks:** Our method addresses the domain generalization problem, with the primary goal of isolating content factors from style factors in the latent space of observations, utilizing content factors for label prediction. In contrast, HVAE focuses more on learning data generation models to generate new observations. (2) **Data generation process:** We specify a specific data generation process from the high-level space to the middle-level space, where the mapping functions of content factors are influenced by the label while the mapping functions of style factors are affected by both the domain index and label. (3) **Distribution assumption on the style variables in the middle level:** We introduce an assumption regarding the data distributions of style latent variables in the middle level, a key for identifying these factors and facilitating the isolation of content factors.
>
> We will incorporate this detailed discussion in our revision to provide a clearer understanding of the distinctions and connections between our method and the Hierarchical VAE.

---

> ### Author Response · Authors · 2023-11-18
> **Response to reviewer Vftx (part 2/2)**
>
> **[Q2. I understand learning a separated latent space for different data features can be beneficial to learning a smoother model manifold. But how does this model improve the performance of DG? The author mentioned that "The key to achieving DG based on our model is recovering the distribution of content factors and isolating them from style factors." An additional and detailed explanation would be good.]**
>
> Thanks for your suggestion. To address your concerns, we believe it's crucial to provide clarification on two key points. Firstly, we aim to explain why our model can achieve DG through the recovery of content factors. Secondly, we aim to highlight how our model enhances DG performance in comparison to traditional methods.
>
> > Why can our model achieve DG through the recovery of content factors?
>
> In the DG problem, we aim to incorporate knowledge from multiple source domains into a single prediction model that could generalize well on unseen testing domains. Usually, there would be distribution shifts between data distribution collected in different domains. To address this challenge, we employ a systematic approach including three key steps: defining a data generation model, establishing the model identifiability conditions, and designing a learning method. **The reason for recovering content factors to achieve DG relies on our data generation model.**
>
> *Data Generation Model:* According to our data generation model, distribution shifts across domains come from variations in the style factors within the middle-level latent space. Importantly, note that the distribution of content factors remains invariant across domains and contains the label information of observations we aim to predict. (Although style factors also contain label information, their variability across domains poses a challenge in developing a stable predictor that relies on them and is applicable to all domains.) Therefore, according to this model, the key to DG lies in isolating the invariant content factors and training a stable predictor based on these features. **When getting observations from a new domain, our model identifies the content factors and utilizes the stable predictor to make accurate label predictions.** Then, we establish identifiability conditions for our proposed model. These conditions guarantee that the content factors can be successfully recovered and isolated from the style factors. We introduce a practical learning method to implement our proposed model effectively.
>
> > But how does this model enhance DG performance in comparison with traditional methods?
>
> Indeed, many previous DG methods assume that labels and non-causal features are independent. Then, these methods try to separate the causal features (label-related) and use them for prediction. However, a significant drawback of these methods emerges: they may include some specific (causal) features whose distributions may change across domains yet are still related to labels, which will lead to instability when applied to unseen domains.
>
> To deal with this case, we propose a novel approach incorporating a two-level latent space to accurately capture the generation of unstable features. Our theoretical framework establishes the identifiability of these latent features. Notably, our model surpasses prior methods by accommodating the interdependence between content and style factors, thereby relaxing the independence assumptions.
>
> We have updated this discussion in our revision to help the audience who may have similar concerns.

---

> > ### Comment · Reviewer_Vftx · 2023-11-22
> >
> > I would like to thank the authors for their response. My questions and concerns are addressed in detail, and I would like to keep my score.

---

### Official Review · Reviewer_XKs5 · 2023-10-31

**Soundness:** 2 fair
**Presentation:** 3 good
**Contribution:** 2 fair
**Rating:** 5
**Confidence:** 3

**Summary:**

This paper aims to recover the label-specific content factors and isolate them from the style ones utilizing a two-level latent space, and consequently learn a stable predictor applicable to all domains. Specifically, they propose a novel data generation process and then exploit a VAE-based framework to achieve the latent variable identifiability based on some assumptions. Theoretical analysis and various experimental results demonstrate the effectiveness of this method.

**Strengths:**

1.This work proposes a two-level latent space that allows for better identifiablility of latent content factors from raw data.

2.This work designs a practical VAE-based framework and also provides sufficient theoretical analysis.

3.Extensive experiments that conducted on both synthetic and real-world datasets show the validity of the proposed framework.

**Weaknesses:**

1.It is common to decouple the raw data into domain-invariant and domain-specific parts in feature disentanglement methods and some causality-based methods. Though this work proposes a two-level latent space to assist the isolation of latent variables, the novelty is still limited.

2.The proposed framework requires the use of domain variables, which are not accessible in some cases, limiting the application of this method.

3.To better show the superiority of this work, it is necessary to compare the experimental results of the proposed method with that of the similar works, such as LaCIM, iMSDA, and the works mentioned in Sec 3.1. Besides, the baselines compared in tables are somewhat outdated.

4.The authors are suggested to evaluate the algorithm on Domainnet dataset to verify its effectiveness on large-scale datasets.

**Questions:**

See Weakness.

---

> ### Author Response · Authors · 2023-11-19
> **Response to Reviewer XKs5 (part 1/2)**
>
> We are grateful for your support and detailed comments! We provide responses to address your comments below.
>
> **[Q1. Novelty.]**  Agree with your observation that our work doesn’t directly contribute to these two aspects: (1) the proposition of separating the latent space into content and style parts; (2) the creation of new identifiability theories for nonlinear ICA. Our theoretical findings and the methodology we propose are fundamentally built upon these prior foundations.
>
> In this context, it’s crucial to re-emphasize that our contributions mainly include the following three primary aspects: (1) the introduction of a novel two-level latent space that relaxes the independence assumptions between label and style factors, distinguishing it from previous methodologies; (2) the establishment of corresponding identifiability results for our proposed data generation model, primarily drawing upon existing theoretical findings; (3) the introduction of a novel and practical VAE-based training method to identify the invariant features. In summary, our work provides a substantial contribution to the field by redefining and building upon established principles to tackle specific challenges and drive advancements in a focused direction.
>
> **[Q2. The use of domain variable.]**
>
> In our paper, we address the multi-source domain generalization (DG) problem—a standard scenario where the learning process can use training data from various domains with domain indexes. In our method, we assume that the domain variable can also affect the style factors across different domains. We rely solely on domain indexes to distinguish between generation models for each domain, eliminating the need for additional information.
>
> A more intricate challenge arises in the single-source DG setting, where no domain labels are available, and the model must learn from data in just one domain, aiming to create a universal predictor applicable to unseen domains.
>
> Theoretically, identifying invariant features proves to be a formidable task, even in the context of multi-source DG, let alone in the more complex single-source DG scenario. In our current paper, we concentrate on the multi-source DG aspect, leaving the single-source DG challenge as a direction for future exploration. We will make our objective more clear and add this discussion to the revision.

---

> ### Author Response · Authors · 2023-11-19
> **Response to Reviewer XKs5 (part 2/2)**
>
> **[Q3. More baselines.]**
>
> > Comparison with LaCIM and iMSDA
>
> iMSDA is designed to address the multi-domain adaptation problem, where it leverages the marginal observations $\mathbf{x}$ from the testing domain to achieve model identifiability. In contrast, DG only relies on data from training domains to develop a preditor applied to the testing data.
>
> We compare the results with LaCIM on the simulation data and Colored MNIST data since they only share the official codes for these datasets.  We did not compare with LaCIM on other datasets due to the unavailability of LaCIM's results on these specific datasets. Additionally, LaCIM has not provided some training details, making it difficult for us to replicate the methods. However, we still compare our method with many recent important algorithms to show its effectiveness.
>
> > Comparison with more recent methods.
>
> According to your suggestion, we have incorporated two recently published methods, Wang-ICLR2023 [1] and Dayal-NeurIPS2023 [2], for a comprehensive comparison. Wang-ICLR2023 proposes a spurious-free balanced distribution to produce an optimal classifier for DG, and Dayal-NeurIPS2023 also tries to learn domain-invariant features by a margin loss-based discrepancy metric. The results are shown below:
>
> Results on PACS.
>
> | Method            | P   |A | C |S | Avg.|
> |-------------------|----|------|------|------|-----|
> | Wang-ICLR2023     |97.1±0.4|87.8±0.8|81.0±0.1|81.1±0.8|86.7|
> | Dayal-NeurIPS2023 |97.7±0.3|87.8±0.5|82.2±0.6|78.3±0.4|86.5|
> | Ours              |97.7±0.4|88.7±0.3|80.6±0.8|82.6±0.4|87.4|
>
> Results on Office-Home.
>
> | Method            | P   |A | C |R | Avg.|
> |-------------------|----|------|------|------|-----|
> | Wang-ICLR2023     |77.6±0.3|65.6±0.6|56.5±0.6|78.8±0.7|69.6|
> | Dayal-NeurIPS2023 |78.4±0.3|67.6±0.2|54.1±0.6|80.3±0.5|70.1|
> | Ours              |78.4±0.6|64.8±0.7|56.1±0.3|79.8±0.3|69.8|
>
> Based on the above results, we can see that our method performs better than the two compared methods when evaluated on the PACS dataset. In the context of the Office-Home dataset, our method outperforms Wang-ICLR2023, although it exhibits a slight underperformance when compared to the results achieved by Dayal-NeurIPS2023.
>
> These results have been updated in the revision.
>
> **[Q4. Results on Domainnet.]** Thanks for your kind suggestion. We are currently in the process of testing our methods and running more baseline methods on additional datasets to verify the effectiveness of our method. Because DomainNet is a large dataset, containing 345 categories and 600,000 images, and since we need to perform a search for hyperparameters of our model, accomplishing this within a short time is quite challenging. We will update these results in the final version.
>
> [1] [Wang, Xinyi, Michael Saxon, Jiachen Li, Hongyang Zhang, Kun Zhang, and William Yang Wang. Causal Balancing for Domain Generalization. In The Eleventh International Conference on Learning Representations. 2023.](https://openreview.net/pdf?id=F91SROvVJ_6)
>
> [2] [Dayal, Aveen, K. B. Vimal, Linga Reddy Cenkeramaddi, C. Krishna Mohan, Abhinav Kumar, and Vineeth N. Balasubramanian. MADG: Margin-based Adversarial Learning for Domain Generalization. In Thirty-seventh Conference on Neural Information Processing Systems. 2023.](https://arxiv.org/pdf/2311.08503.pdf)

---

### Official Review · Reviewer_f1Ds · 2023-11-01

**Soundness:** 3 good
**Presentation:** 3 good
**Contribution:** 2 fair
**Rating:** 6
**Confidence:** 3

**Summary:**

This paper proposes a novel approach for domain generalization by introducing a two-level latent variable model.
The key idea is to partition the latent space into invariant content factors and variant style factors across domains. Specifically, the high-level latent space consists of content and style variables.
The content variables capture label-related information while the style variables represent domain-specific information.
To achieve identifiability, the model introduces a middle-level latent space with the same partition structure.
The middle-level content factors are derived from the high-level content factors via label-specific functions.
Similarly, the middle-level style factors are obtained by applying component-wise monotonic functions to the high-level style factors, which depend on the label, domain, and variable index.
The observation is then generated by applying a mixing function on the middle-level latent factors, which is shared across domains.
The key theoretical contribution is providing sufficient conditions to achieve identifiability and isolation of the content factors from the style factors. This relies on assuming the style factors follow an exponential family distribution conditioned on label and domain, plus a domain variability assumption.
Under these assumptions, the authors prove the content factors can be block-identified and style factors can be linearly identified.
Based on the theoretical results, the paper proposes a practical learning algorithm using a VAE framework.
The VAE encoder estimates the posterior of the latent factors.
Normalizing flows are used to transform between the high-level and middle-level latent variables.
An invariant classifier is trained solely on the recovered content factors.
Experiments on synthetic and real-world image datasets demonstrate the approach can effectively identify the latent factors and that training on just the content factors improves domain generalization performance.

**Strengths:**

- This paper made contributions: 1) A novel identifiable latent variable model with content/style partition 2) Sufficient conditions for identifiability and isolation of content factors 3) A practical learning algorithm based on VAEs and normalizing flows 4) Strong empirical performance on domain generalization tasks. The proposed approach offers a promising way to learn invariant representations for generalizable models. This paper proposes a novel two-level latent variable model with content/style partitioning to achieve domain generalization. This framework allows dependence between factors while still enabling identifiability. This work introduces sufficient conditions for identifiability and isolation of content factors based on exponential family priors and domain variability assumptions, combining VAEs and normalizing flows in a new way to estimate latent factors for domain generalization.
- Provides thorough theoretical analysis and identifiability guarantees for the proposed model.
- Learning invariant representations is an important open problem for building generalizable ML models. Methodology could be applied to other domain generalization areas beyond image classification.

**Weaknesses:**

- The method relies on specific assumptions about the latent variable distributions which may not hold universally. For example, the exponential family prior and domain variability assumptions.
- The theoretical analysis requires infinite data and may not provide guarantees for small sample sizes. More analysis of finite sample behavior would be useful.
- The model structure imposes some limitations, like only allowing dependence between factors through the label, but more complex relationships may exist in real datasets.
- While state-of-the-art results are shown, the gains are incremental. More significant jumps in performance may be needed to drive adoption.

**Questions:**

Please see above.

---

> ### Author Response · Authors · 2023-11-19
> **Response to Reviewer f1Ds (part 1/3)**
>
> We are grateful for your support and detailed comments! We provide responses to address your comments below.
>
> **[Q1. The method relies on specific assumptions about the latent variable distributions which may not hold universally. For example, the exponential family prior and domain variability assumptions.]**
>
> Thank you for your insightful suggestion! We fully acknowledge that making assumptions on prior and domain variability can be challenging and maybe impractical to handle real-world data. In the context of the domain generalization (DG) problem considered in our paper, identifying and isolating content factors from style factors within the latent space emerges as a key method, which contributes to the learning of a robust predictor capable of safely generalizing data from previously unseen distributions.
>
> However, the pursuit of model identifiability, particularly with latent variables, poses a formidable and ongoing challenge. Traditional VAE approaches often hinge on the mild assumption that the data distribution of latent variables adheres to a normal distribution, making the problem unsolvable [1]. In contrast, nonlinear ICA methods introduce a different paradigm by constraining the prior of latent variables to the exponential family, where sufficient statistics are controlled by auxiliary information, such as time index or domain index. This is a trade-off between the model assumptions and model identifiability.
>
> In alignment with the Nonlinear ICA techniques [2,3], our model also adopts a similar prior assumption on latent variables to ensure identifiability and leverage the label information and domain details to control the sufficient statistics of the prior distribution. Nevertheless, it is important to note that in scenarios with a large number of latent variables, our model may encounter challenges in providing a robust guarantee for identifying these variables. Moreover, if the style factors are not affected by the label of observations, our method may also fail to identify the latent variables.
>
> We have included all this discussion in our revision to make the assumptions clear to audiences and clarify the constraints of our current model.
>
> **[Q2. The theoretical analysis requires infinite data and may not provide guarantees for small sample sizes. More analysis of finite sample behavior would be useful.]**
>
> Agree with you that small sample sizes pose a huge challenge and prevent the Nonlinear ICA methods from achieving model identfiability without further assumptions. Conversely, our theoretical framework assumes infinite data and posits a universal function approximator, both of which may be unattainable in practical learning scenarios. In such real-world settings, the analysis of estimation errors becomes crucial, yielding an upper bound that is contingent on the complexity and smoothness of the demixing function, along with the sample size.
>
> In practical terms, it is intuitive that learning a more intricate demixing necessitates a larger volume of data. Simultaneously, the lack of smoothness in the demixing function exacerbates the complexity of the estimation process [4]. However, to our knowledge, there is only one paper [5] that delves into the finite sample convergence of Nonlinear ICA methods. Notably, their focus remains confined to contrast-based learning methods—a facet that poses a non-trivial challenge when extending the findings to our methodology.
>
> As you suggested, we have added a discussion in our revision and leave this analysis as future work.

---

> ### Author Response · Authors · 2023-11-19
> **Response to Reviewer f1Ds (part 2/3)**
>
> **[Q3. The model structure imposes some limitations, like only allowing dependence between factors through the label, but more complex relationships may exist in real datasets.]**
>
> Indeed, many previous DG methods assume that labels and non-causal features are independent. Then, these methods try to separate the causal features (label-related) and use them for prediction. However, a significant drawback of these methods emerges: they may include some specific (causal) features whose distributions may change across domains yet are still related to labels, which will lead to instability when applied to unseen domains. To deal with this case, we introduce a novel two-level latent space to depict the generation of such unstable features and establish theoretical results for the identifiability of latent features. We also develop a practical method based on VAEs to effectively disentangle these features from the causal ones.
>
> We acknowledge that more complex relationships may exist in real datasets, such as image datasets, where the underlying data generation model may violate the assumptions in our paper. We are leaving these more complex cases as future work.
>
> **[Q4. While state-of-the-art results are shown, the gains are incremental. More significant jumps in performance may be needed to drive adoption.]**
>
> We appreciate your acknowledgment of the state-of-the-art performance. To verify the effectiveness of our proposed method, we have incorporated two recently published methods, Wang-ICLR2023 [6] and Dayal-NeurIPS2023 [7], for a comprehensive comparison. Wang-ICLR2023 proposes a spurious-free balanced distribution to produce an optimal classifier for DG, and Dayal-NeurIPS2023 also tries to learn domain-invariant features by a margin loss-based discrepancy metric. The results are shown below:
>
> Results on PACS.
>
> | Method            | P   |A | C |S | Avg.|
> |-------------------|----|------|------|------|-----|
> | Wang-ICLR2023     |97.1±0.4|87.8±0.8|81.0±0.1|81.1±0.8|86.7|
> | Dayal-NeurIPS2023 |97.7±0.3|87.8±0.5|82.2±0.6|78.3±0.4|86.5|
> | Ours              |97.7±0.4|88.7±0.3|80.6±0.8|82.6±0.4|87.4|
>
> Results on Office-Home.
>
> | Method            | P   |A | C |R | Avg.|
> |-------------------|----|------|------|------|-----|
> | Wang-ICLR2023     |77.6±0.3|65.6±0.6|56.5±0.6|78.8±0.7|69.6|
> | Dayal-NeurIPS2023 |78.4±0.3|67.6±0.2|54.1±0.6|80.3±0.5|70.1|
> | Ours              |78.4±0.6|64.8±0.7|56.1±0.3|79.8±0.3|69.8|
>
> Based on the above results, we can see that our method performs better than the two compared methods when evaluated on the PACS dataset. In the context of the Office-Home dataset, our method outperforms Wang-ICLR2023, although it exhibits a slight underperformance when compared to the results achieved by Dayal-NeurIPS2023. Moreover, we believe that there is some gap between our assumptions and the real image datasets which leads to the incremental gains of our proposed method. Moreover, as you acknowledged, our main contributions rely on 1) A novel identifiable latent variable model with content/style partition 2) Sufficient conditions for identifiability and isolation of content factors 3) A practical learning algorithm based on VAEs and normalizing flows 4) Strong empirical performance on domain generalization tasks. The proposed approach offers a promising way to learn invariant representations for generalizable models. Therefore, we still believe that our method makes contributions to give a new modeling way with the theoretical guarantee to deal with the DG problem.

---

> ### Author Response · Authors · 2023-11-19
> **Response to Reviewer f1Ds (part 3/3)**
>
> [1] [Aapo Hyvärinen and Petteri Pajunen. Nonlinear independent component analysis: Existence and uniqueness results. Neural networks, 12(3):429–439, 1999.](https://www.sciencedirect.com/science/article/abs/pii/S0893608098001403)
>
> [2] [Ilyes Khemakhem, Diederik Kingma, Ricardo Monti, and Aapo Hyvarinen. Variational autoencoders and nonlinear ica: A unifying framework. In International Conference on Artificial Intelligence and Statistics, pp. 2207–2217. PMLR, 2020a.](https://arxiv.org/abs/1907.04809)
>
> [3] [Ilyes Khemakhem, Ricardo Monti, Diederik Kingma, and Aapo Hyvarinen. Ice-beem: Identifiable conditional energy-based deep models based on nonlinear ica. Advances in Neural Information Processing Systems, 33:12768–12778, 2020b.](https://arxiv.org/abs/2002.11537)
>
> [4] [Hyvärinen A, Khemakhem I, Morioka H. Nonlinear independent component analysis for principled disentanglement in unsupervised deep learning. Patterns. 2023 Oct 13;4(10).](https://www.sciencedirect.com/science/article/pii/S2666389923002234)
>
> [5] [Lyu, Qi, and Xiao Fu. On Finite-Sample Identifiability of Contrastive Learning-Based Nonlinear Independent Component Analysis. In International Conference on Machine Learning, pp. 14582-14600. PMLR, 2022.](https://arxiv.org/abs/2206.06593)
>
> [6] [Wang, Xinyi, Michael Saxon, Jiachen Li, Hongyang Zhang, Kun Zhang, and William Yang Wang. Causal Balancing for Domain Generalization. In The Eleventh International Conference on Learning Representations. 2023.](https://openreview.net/pdf?id=F91SROvVJ_6)
>
> [7] [Dayal, Aveen, K. B. Vimal, Linga Reddy Cenkeramaddi, C. Krishna Mohan, Abhinav Kumar, and Vineeth N. Balasubramanian. MADG: Margin-based Adversarial Learning for Domain Generalization. In Thirty-seventh Conference on Neural Information Processing Systems. 2023.](https://arxiv.org/pdf/2311.08503.pdf)

---

### Official Review · Reviewer_2mKD · 2023-11-03

**Soundness:** 3 good
**Presentation:** 2 fair
**Contribution:** 3 good
**Rating:** 6
**Confidence:** 3

**Summary:**

This paper proposes two-level latent variables for improving domain generalization. The key idea is to employ other $z_c$ and $z_s$ to  recover the distribution of content factors and isolating them from style factors.

**Strengths:**

- The idea is innovative and interesting.

- The experimental results are convincing.

**Weaknesses:**

- The notions used in this paper are confusing, making it hard to read. The authors can simplify the notions in formal setup because it seems that the paper only uses a few of them. Moreover, in Eq. (1), what is the meaning of the distribution inside $\phi(.,.)$.

- The generative process or data generation model also confuses me. It is not clear why $f_y(\hat{z_c})$ can return $z_{c_1}, z_{c_2}, z_{c_3}$. Also, the same question is for the style branch. What are $\hat{z_{s_1}}, \hat{z_{s_2}}, \hat{z_{s_3}}$? Are they the styles of the domains?

- Eq. (7) is also hard to interpret to me. As far as I understand, in the first level you have a single variable $\hat{z_s}$ for the style and using the map $f_{e,y,i}$ to transform it to $p(z_s \mid y,e)$. However, why do you need the index $i$ here?

**Questions:**

Please answer my questions in the weakness session. Moreover, how do you design $f_y$ and $f_{e,y,i}$ using distinct flow-based architecture to incorporate the information of e, y, i? I am happy to increase my score if the authors can resolve my unclear points.

---

> ### Author Response · Authors · 2023-11-19
> **Response to reviewer 2mKD (part 1/3)**
>
> Thank you very much for your insightful comments. Your suggestions have been invaluable in clarifying some confusing representations and further enhancing the quality of our paper.
>
> **[Q1. The notions used in this paper are confusing, making it hard to read. The authors can simplify the notions in formal setup because it seems that the paper only uses a few of them. Moreover, in Eq. (1), what is the meaning of the distribution inside $\phi(\cdot,\cdot)$]?**
>
> > The authors can simplify the notions in formal setup because it seems that the paper only uses a few of them.
>
> Thanks for this suggestion. We have simplified the notions in the formal setup part of our revision.
>
> > what is the meaning of the distribution inside $\phi(\cdot,\cdot)$?
>
> $\hat{P}^T_{\mathbf{x}}$ represents the **empirical distribution** associated with observations $\mathbf{x}$ in the testing data. Notice that this empirical distribution is obtainable in the domain generalization (DG) setting during the testing phase.
>
> For the DG problem, the training set comprises data from different distributions across several domains. The objective is to develop an estimator to predict labels when presented with observations from the testing set, which is drawn from an unseen distribution. To define the estimator $\phi:=\mathcal{P}_{\mathcal{X}}\times\mathcal{X}\rightarrow \mathbb{R}$ completely, we incorporate both the empirical distribution and data within the input space following the same way as in [1]. When $\phi$ is shared across different domains, the distribution term becomes dispensable. Consequently, in the revised version, we have removed this term, as the estimator is shared for data from all domains in our method. Additionally, we have excluded the definition of the empirical distribution, as it is not subsequently referenced in this paper.

---

> ### Author Response · Authors · 2023-11-19
> **Response to reviewer 2mKD (part 2/3)**
>
> **[Q2. The generative process or data generation model also confuses me. It is not clear why $f _y(\hat{z} _c)$ can return  $z _{c _1},z _{c _2},z _{c _3}$. Also, the same question is for the style branch. What are $\hat{z} _{c_1}, \hat{z} _{c_2}, \hat{z} _{c_3}$? Are they the styles of the domains?]**
>
> We acknowledge that the previous Figure 1 was confusing due to inappropriate representation, potentially confusing your understanding. We have replotted this figure in the revision to make it clear.
>
> To address your concern, let us focus on the detailed example in Figure 1, in which **both the content factors and style factors in the high-level space are characterized by three dimensions**. Specifically, denoting the content facotrs as $\hat{\mathbf{z}} _c = \\{\hat{\mathbf{z}} _{c _1},\hat{\mathbf{z}} _{c _2},\hat{\mathbf{z}} _{c _3}\\}$, and the style factors as $\hat{\mathbf{z}} _s = \\{\hat{\mathbf{z}} _{s _1},\hat{\mathbf{z}} _{s _2},\hat{\mathbf{z}} _{s _3}\\}$. Similarly, the latent variables within the middle-level space are denoted as $\mathbf{z} _c = \\{\mathbf{z} _{c _1},\mathbf{z} _{c _2},\mathbf{z} _{c _3}\\}$, $\mathbf{z} _s = \\{\mathbf{z} _{s _1},\mathbf{z} _{s _2},\mathbf{z} _{s _3}\\}$. Following the general assumption of VAE, we assume that all variables in the high-level space, $\\{\hat{\mathbf{z}} _{c _1}, \hat{\mathbf{z}} _{c _2}, \hat{\mathbf{z}} _{c _3}, \hat{\mathbf{z}} _{s _1}, \hat{\mathbf{z}} _{s _2}, \hat{\mathbf{z}} _{s _3}\\}$, are independent. To elaborate further, let us consider a scenario where we have $3$ distinct classes denoted as $\\{ y _1,y _2,y _3\\}$ and $2$ different domains denoted as $\\{e _1, e _2\\}$ in the training set.
>
> > Why $f^* _y(\hat{\mathbf{z}} _c)$ can return $\mathbf{z} _{c _1}, \mathbf{z} _{c _2}, \mathbf{z} _{c _3}$?
>
> In the previous Figure, we did not divide $\hat{\mathbf{z}} _c$ into $\{\hat{\mathbf{z}} _{c _1},\hat{\mathbf{z}} _{c _2},\hat{\mathbf{z}} _{c _3}\}$, which may lead to your confusion.
>
> In this particular example, we have a total of three functions denoted $f^* _{y _1}$, $f^* _{y _2}$, and $f^* _{y _3}$. Each of these functions takes a three-dimensional input and produces a three-dimensional output, mapping from $\hat{\mathbf{z}} _c = \\{\hat{\mathbf{z}} _{c _1}, \hat{\mathbf{z}} _{c _2}, \hat{\mathbf{z}} _{c _3}\\}$ to $\mathbf{z} _c = \\{\mathbf{z} _{c _1}, \mathbf{z} _{c _2}, \mathbf{z} _{c _3}\\}$. It's worth noting that the selection of these functions is solely dependent on the label $y$.
>
> > What are $\hat{\mathbf{z}} _{s _1}, \hat{\mathbf{z}} _{s _2}, \hat{\mathbf{z}} _{s _3}$? Are they the styles of the domains?
>
> $\\{\hat{\mathbf{z}} _{s _1}, \hat{\mathbf{z}} _{s _2}, \hat{\mathbf{z}} _{s _3}\\}$ are high level style factors. However, their distributions are invariant across domains.
>
> Let's consider the image dataset as an example. The middle-level style factors denoted as $\\{\mathbf{z} _{s _1},\mathbf{z} _{s _2},\mathbf{z} _{s _3}\\}$, can be associated with various types of backgrounds present in the images, such as sky, forest, or lake scenes. In this context, the factors $\\{\hat{\mathbf{z}} _{c _1}, \hat{\mathbf{z}} _{c _2}, \hat{\mathbf{z}} _{c _3}\\}$ can be interpreted as a generic background pattern. This pattern is versatile and can be easily transformed into a domain-specific image background, influenced by distinct $f^\\# _{y,e}(\cdot)$ functions across different domains. Notice that the distributions of middle-level style factors $\\{\mathbf{z} _{s _1}, \mathbf{z} _{s _2}, \mathbf{z} _{s _3}\\}$ are different across different domains.
>
> > Why $f^\\# _{y,e}(\hat{\mathbf{z}} _s)$ can return $\mathbf{z} _{s _1},\mathbf{z} _{s _2}, \mathbf{z} _{s _3}$?
>
> Similarly, we have in total $n_y\times n_e=3\times 2=6$ distinct functions for $f^\\# _{y,e}$, saying $\\{f^\\# _{y _1,e _1}, f^\\# _{y _1,e _2}, f^\\# _{y _2,e _1}, f^\\# _{y _2,e _2}, f^\\# _{y _3,e _1}, f^\\# _{y _3,e _2}\\}$, each of which is a component-wise monotonic function and takes a three-dimensional input and produces a three-dimensional output, mapping from $\hat{\mathbf{z}} _s = \\{\hat{\mathbf{z}} _{s _1}, \hat{\mathbf{z}} _{s _2}, \hat{\mathbf{z}} _{s _3}\\}$ to $\mathbf{z}_s = \\{\mathbf{z} _{s _1}, \mathbf{z} _{s _2}, \mathbf{z} _{s _3}\\}$. The choice of functions is dependent both on the label $y$ and domain $e$.

---

> ### Author Response · Authors · 2023-11-19
> **Response to reviewer 2mKD (part 3/3)**
>
> **[Q3. Eq. (7) is also hard to interpret to me. As far as I understand, in the first level you have a single variable $\hat{z} _s$  for the style and using the map  $f _{e,y,i}$  to transform it to  $p(z_s|y,e)$. However, why do you need the index  $i$  here?]**
>
> > in the first level you have a single variable $\hat{\mathbf{z}} _s$ and using the map  $f^\\# _{y,e}$  to transform it to  $p(z_s|y,e)$.
>
> As clarified earlier, we have a total of $n_s$ style variables, denoted as $\hat{\mathbf{z}} _s = \\{\hat{\mathbf{z}} _{s _1}, \cdots, \hat{\mathbf{z}} _{s _{n_s}}\\}$ instead of only one single variable in the high-level space, and also $n_s$ style variables, denoted as $\mathbf{z} _s = \\{\mathbf{z} _{s _1}, \cdots, \mathbf{z} _{s _{n _s}}\\}$ in the middle-level space.
>
> We assume that the high-level style factors are mutually independent (a common assumption in VAE) and invariant across different domains. The key to distribution shifts across domains comes from the variations in the distributions of middle-level style factors $\mathbf{z} _s = \\{\mathbf{z} _{s _1}, \cdots, \mathbf{z} _{s _{n _s}}\\}$, as the transformation function is dependent on the choice of domain.
>
> > However, why do you need the index $i$ here?
>
> In Eq.(7), we constrain the distributions of the middle-level style factors, which is an important assumption for the identifiability of the style factors. We use $i$ from $1$ to $n_s$ to index the middle-level style factors.
>
> **[Q4. Moreover, how do you design $f_y$ and $f_{e,y,i}$ using distinct flow-based architecture to incorporate the information of $e, y, i$?]**
>
> There are two ways to incorporate the information of $y$ or $e,y$ to design distinct flows. Let us take $e,y$ for example.
>
> The first method is directly employing independent flows for each function. For example, when learning $\\{f^\\# _{y _1,e _1},f^\\# _{y _1,e _2},f^\\# _{y _2,e _1},f^\\# _{y _2,e _2},f^\\# _{y _3,e _1},f^\\# _{y _3,e _2}\\}$, we can directly utilize $6$ independent flows for each corresponding function. This method is employed in our simulation experiment.
>
> However, the first method poses challenges in terms of computational complexity, particularly when dealing with numerous functions. In real image datasets, we follow the method in [2], where we learn a distinct embedding (a constant vector, in our case, a size of $1\times1024$) for each combination of $(y, e)$. Subsequently, we integrate this embedding into the model to influence certain parameters of the flow model. This way can significantly mitigate computational complexity.
>
> [1] [Blanchard, Gilles, Gyemin Lee, and Clayton Scott. Generalizing from several related classification tasks to a new unlabeled sample. Advances in neural information processing systems 24 (2011).](https://proceedings.neurips.cc/paper_files/paper/2011/file/b571ecea16a9824023ee1af16897a582-Paper.pdf)
>
> [2] [Kong, Lingjing, Shaoan Xie, Weiran Yao, Yujia Zheng, Guangyi Chen, Petar Stojanov, Victor Akinwande, and Kun Zhang. Partial disentanglement for domain adaptation. In International Conference on Machine Learning, pp. 11455-11472. PMLR, 2022.](https://proceedings.mlr.press/v162/kong22a.html)

---

> > ### Comment · Reviewer_2mKD · 2023-11-22
> > **Response to the authors**
> >
> > The authors have clarified my concerns and modified the paper accordingly. So I decide to increase my score.

---

### Meta-Review · Area_Chair_JW29 · 2023-12-04

**Metareview:**

The paper proposes a method for domain generalization based on isolating content factors that are stable predictors under domain shifts. This is embedded into a VAE framework, leading to a practical method. The reviewers found the paper innovative and the experiments well-executed (although with marginal improvements) and appreciated the identifiable latent variable model approach. The main limitations of the papers were the clarity in the notation, which improved during the rebuttal, and the overall novelty.

The paper is borderline, and ultimately, I decided to reject it. The main reasons are:

1) I would like a more extensive discussion on the proof technique. It seems like a direct application of Khemakhem's proof, with the label and environment pair taking the place of the auxiliary variable. In fact, step 1 and 2 in the appendix are almost ad verbatim copies of that paper. I would recommend refactoring them out as lemmas, and crediting the proof (no need to rewrite the proof if it's identical). With this, it is very hard to tell whether this is a special case of their theory or not.

2) I find the experiments very limited, only choosing two data sets from the DomainBed suite (also with marginal improvements). It is hard to draw strong conclusions from these. Minor: The ColorMNIST results seem to miss the OOD test accuracy and only report the in-distribution numbers.

**Justification For Why Not Higher Score:**

Lack of clarity in the novelty of the theoretical contribution and limited experimental results.

**Justification For Why Not Lower Score:**

N/A

---

### Decision · Program_Chairs · 2024-01-16

Reject